# Thin film lithium niobate on sapphire for integrated mid-infrared modulator

Pierre Didier [1] ✉, Prakhar Jain[1], Mathieu Bertrand [2], Jost Kellner[1], Oliver Pitz [1], Zhecheng Dai[3], Tristan Kuttner [1], Mattias Beck [2], Baile Chen [3], Jérôme Faist [2] & Rachel Grange [1]

The mid-infrared spectrum, spanning from 3 to 14 μm, holds great promise for molecular spectroscopy and free space optical communication, benefiting from strong molecular absorption and reduced atmospheric attenuation. While progress in MIR photonics has accelerated due to improved sources and detectors, integrated low-loss, high -performance modulators remain limited. In order to address this gap, we demonstrate a broadband, high speed lithium niobate on sapphire Mach Zehnder electro optic modulator operating from 3.95 to 4.5 μm. The device shows a 3 dB bandwidth above 20 GHz, 17 dB extinction ratio, and $V_\pi L = 22$ V · cm, with optical output power at the half milliwatt level. We demonstrate 10 Gbit s$^{-1}$ data transmission and a 70 GHz broad frequency comb, uniquely combining integration, low propagation loss, extinction ratio and high-speed operation.

Mid-infrared (MIR) photonics requires the development of highly efficient platforms to fully exploit the unique advantages of this spectral region. The MIR spectrum, spanning 3–14 μm, offers significant benefits for free-space optical applications[1,2], including high atmospheric transparency, reduced scattering from micrometric aerosols, and enhanced resilience to atmospheric turbulence[3–5]. Furthermore, many environmentally relevant molecules exhibit strong absorption features in the MIR that are up to an order of magnitude stronger than in the near-infrared, enabling more precise and sensitive detection[6,7]. In this context, efficient electro-optic modulators play a crucial role in both domains. External modulators, in particular, enable high modulation depths and can handle significantly higher optical power than directly modulated sources, making them highly advantageous for achieving strong, high-quality modulated signals in the mid-infrared[8]. The MIR spectral range also offers several benefits for spectroscopy, for which precise control over the phase and amplitude of mid-infrared light is required. Amplitude modulation enables lock-in detection to improve sensitivity and suppress background signals[9,10], and it can also be used to probe very fast phenomenon[11] and delicate samples[12]. For example, Tamamitsu et al. use a 0.1 ns (10 GHz) MIR pulse to confine heat diffusion and enhance photothermal efficiency,

although it is currently generated by a bulky OPO system. Phase modulation, on the other hand, enables wavelength modulation spectroscopy[12] and the generation of frequency-comb light sources[13], providing powerful tools for high-resolution and broadband spectroscopy. Achieving such performance in the MIR requires modulators that combine strong electro-optic efficiency, low propagation loss, and broadband operation, which remain challenging due to material and fabrication limitations in this wavelength range.

There have been many efforts towards amplitude modulation in the MIR, each with their own advantages and drawbacks. The most common demonstrations have been with direct modulation of quantum or interband cascade lasers (QCLs/ICLs)[14–16]. QCLs operate most reliably in the long-wave infrared (3.8–14 μm) range. While RF packaging requirements can introduce thermal constraints, QCLs can be directly modulated at high speeds. Promising optical 3 dB bandwidths up to 3 GHz have been demonstrated[17,18], but the modulation depth remains relatively low (approximately 10 dB) due to strong non-linearities at high drive currents, along with the need for large current swings and high electrical power. ICL-based schemes have demonstrated gigahertz-scale performance[19], benefiting from less demanding thermal management requirements, although they do suffer from

[1]ETH Zurich, Department of Physics, Institute for Quantum Electronics, Optical Nanomaterial Group, Zurich, Switzerland. [2]ETH Zurich, Department of Physics, Institute for Quantum Electronics, Quantum Optoelectronics Group, Zurich, Switzerland. [3]School of Information Science and Technology, ShanghaiTech University, Shanghai, China. ✉e-mail: pdidier@phys.ethz.ch

limited output power. Moreover, direct modulation of QCLs/ICLs inherently causes simultaneous amplitude and phase modulation, limiting their use in some applications. Alternative approaches, such as external free-space Stark-effect modulators, have achieved comparable bandwidths[20–22] and extinction ratios up to 3 dB[23]. These devices are particularly attractive because they rely on a refractive index modulation mechanism rather than carrier absorption, enabling easier free-space coupling and potentially lower insertion losses. However, their scalability remains limited due to the high alignment sensitivity inherent to free-space configurations. In the integrated domain, III-V-based and group-IV carrier-injection or depletion modulators[24] can operate at longer wavelengths (up to 8 μm) but generally exhibit high optical losses. Despite their impressively low voltage-length product ($V_\pi L$), these devices suffer from significant free-carrier absorption. Recent demonstrations include implementations on Ge-on-Si platforms[24] and silicon-on-insulator structures[25], which exhibit relatively high propagation losses due to lattice-mismatch defects between Ge and Si, and absorption in the silicon dioxide substrate, respectively. A newer technology relies on graded-index Ge-on-Si designs[26], which minimize defects and thus reduce losses, but still suffer from shallow modulation depths despite being very promising in terms of wavelength accessibility. Hybrid Silicon/ Lithium Niobate transfer-printed modulators[27,28] have extended lithium niobate operation into the mid-infrared, demonstrating good performance near 3.8 μm and showing promise for integrated spectroscopic sensing. However, fabrication complexity and limited electro-optic overlap still constrain their performance. Finally, lithium niobate on insulator (LNOI) modulators, despite their proven success in the near-infrared, are intrinsically limited to operation below approximately 3.8 μm[29] due to strong absorption in the buried silicon dioxide (SiO$_2$) layer[30]. On the other hand, phase modulation in the mid-infrared has so far been demonstrated in only a few platforms, including optical parametric oscillators (OPOs) based on periodically poled bulk lithium niobate[31,32], Stark-effect devices using intersubband transitions[33,34], and graphene-based structures[35]. While OPOs provide high optical power, they rely on bulky tabletop components and have very high power consumption. As already discussed, carrier depletion or injection modulators can achieve performance approaching a π-phase shift with lower ($V_\pi L$) but suffer high optical losses[24,25].

Lithium niobate (LiNbO$_3$ or LN) presents a promising alternative for the 3–5 μm wavelength range, offering low propagation loss, a wide transparency window (extending beyond 4.5 μm), and a strong second-order non-linearity χ$^2$, with a large Pockels coefficient up to 33 pm/V at 1300 nm[36]. This enables a change of the refractive index via an externally applied electric field. Conventional bulk LN waveguides are based on titanium in-diffusion and suffer from weak mode confinement and large bend radii, which limit both nonlinear efficiency and potential for scalable integration[37]. The advent of the LNOI platform has enabled high-confinement waveguides and compact, high-performance modulators in the C-band reaching bandwidths above 100 GHz with $V_\pi L$ below 1 V[38,39]. However, the LNOI material stack includes a buried SiO$_2$ layer, which absorbs strongly beyond 3.4 μm[30], hence limiting MIR operation. This limitation can be addressed by the lithium niobate on sapphire (LNOS) platform[40], which replaces the SiO$_2$ substrate with sapphire, thereby extending the transparency window up to 4.5 μm. LNOS preserves the electro-optic advantages of lithium niobate while enabling low-loss, high-confinement waveguides in the MIR. Moreover, sapphire offers several attractive properties, including strong acousto-optic effects, low radio frequency loss, and excellent thermal management[41].

We demonstrate an integrated amplitude modulator in the MIR, operating near a wavelength of 4 μm. The modulators presented in this work are based on a traveling-wave Mach-Zehnder interferometer architecture implemented on the LNOS platform. Light is split by a multi-mode interferometer (MMI) into two parallel waveguide arms

positioned within the gap of a ground-signal-ground coplanar microwave line, where tightly confined optical and radio-frequency fields propagate in the same direction. The differential electric field across the arms induces an antisymmetric phase shift via the Pockels effect, enabling amplitude modulation through interference at the output MMI junction. Our modulator exhibits a half-wave voltage-length product $V_\pi L$ of 22 V·cm, while achieving a 3 dB bandwidth of higher than 20 GHz with an extinction ratio of 17.1 dB and an insertion loss of 14.1 dB. To the best of our knowledge, these values of loss, speed, and extinction ratio have not previously been achieved in a single modulator operating in the MIR[42]. We also compare the performance of two device geometries based on different LN film thicknesses, namely 0.9 μm and 1.5 μm. The device demonstrates a strong modulation depth and excellent signal integrity at high frequencies, delivering an output optical power exceeding a usable half milliwatt. We demonstrated operation up to 4.5 μm, although increasing propagation loss limits performance at longer wavelengths. In this work, we present high-quality fabrication on the LNOS platform, achieving low-loss MIR modulators. The process is particularly challenging due to the sapphire substrate's transparency and insulating nature, which hinder charge dissipation and alignment, as well as the need for deep lithium niobate etching (>400 nm) to ensure strong optical confinement at longer MIR wavelengths. These results represent an advancement in integrated mid-infrared photonics towards new capabilities in non-linear mid-infrared optics, coherent sensing, high-resolution spectroscopy, and free-space optical communication.

## Results
### Fabrication and design of the lithium niobate on sapphire platform

A schematic of the device layout is shown in Fig. 1a. The chips were fabricated from commercially available 0.9 μm-- and 1.5 μm-thin film x-cut LNOS wafers. Photonic waveguides were patterned using electron-beam lithography and etched via argon-based reactive ion etching. The fabricated waveguides feature a top width of 4 μm with an etch depth of 400 nm for the 0.9 μm-thick film, and a top width of 2.5 μm with an etch depth of 920 nm for the 1.5 μm-thick film. Notably, our fabrication process allows for deep etching up to 1 μm. The achievable etch depth is limited by fabrication constraints, such as the thickness of the photoresist etch mask. In both cases, the sidewall angle is approximately 65°, resulting from the anisotropic physical etching process[43]. These designs ensure single-mode operation in transverse electrical (TE) polarization at a wavelength of 4 μm, corresponding to the target operating range of the modulator as shown in Fig. 1b. Transverse magnetic modes are supported only for the 1.5-μm-thin film due to the thickness of the LN film. Moreover, with this thickness, the optical mode is significantly more confined within the lithium niobate, with a mode area of 6.2 μm$^2$ compared to 8.89 μm$^2$ for the 0.9 μm film, where the mode also tends to leak into the sapphire substrate. The electro-optic electrodes were patterned using direct laser writing, followed by electron beam evaporation and lift-off of gold. The 900 nm-thick electro-optic electrodes minimize high-frequency losses, enabling high-speed modulation. Thermo-optic electrodes were defined by electron-beam lithography, followed by another evaporation and gold lift-off. They were designed with a thickness of 100 nm to ensure efficient Joule heating while maintaining structural integrity and preventing electrode burnout. The facet of the device was diced, and the waveguide facets were mechanically polished to minimize in- and out-coupling losses, followed by focused ion beam (FIB) milling to further refine the facet quality, as shown in Fig. 1c. A thermo-optic phase shifter (TOPS) is included on one arm of the MZM as seen in Fig. 1d, in order to set the relative phase between the two arms and ensure that the device operates at the quadrature point. The TOPS voltage set point corresponds to the condition where the optical output power of the interferometer is half of its maximum

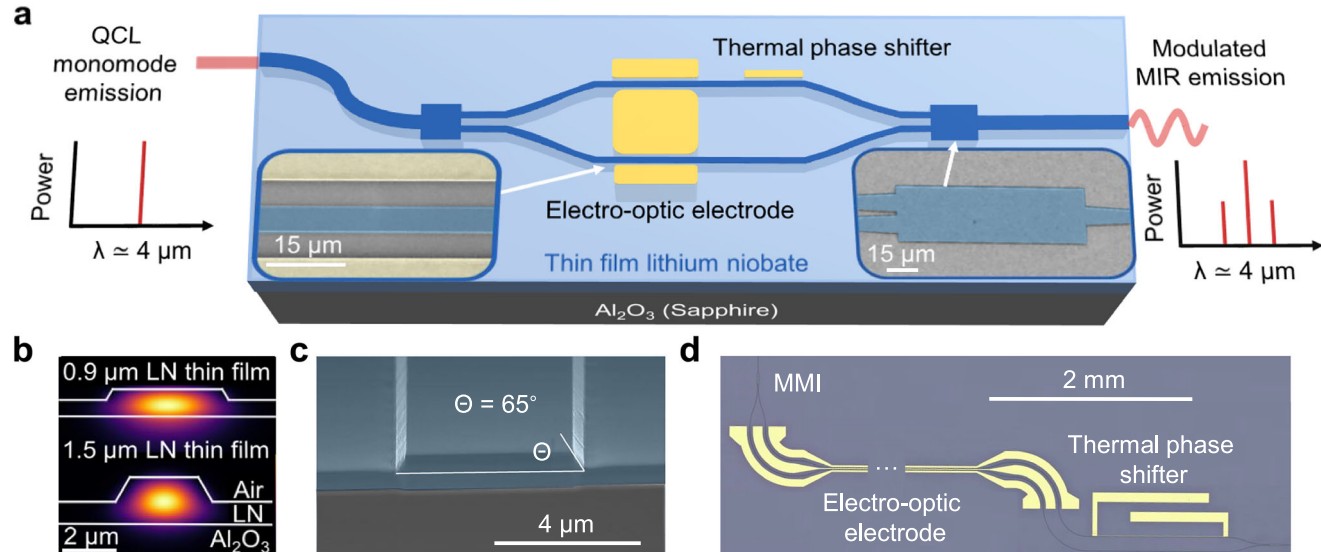

**Fig. 1 | LNOS Mach-Zehnder modulator. a** Schematic of the Mach-Zehnder optical amplitude modulator (MZM). The insets show SEM images of the multi-mode interferometer (bottom right) and electrodes around the waveguide (bottom left). **b** Cross-sectional simulations showing optical mode confinement within lithium niobate thin films of thickness 0.9 μm (top) and 1.5 μm (bottom) at a wavelength of 4 μm. The 0.9 μm film supports only the fundamental transverse electric (TE) mode, while the 1.5 μm film supports both a TE and a transverse magnetic (TM) mode. **c** SEM image of a waveguide facet, mechanically polished and subsequently refined using FIB milling. **d** Optical microscope image of a fabricated device, showing the MMI coupler (top left), electro-optic electrodes (middle), and thermal phase shifter (right).

value, ensuring operation at the quadrature point for optimal electro-optic modulation efficiency. This condition was also verified by observing the modulator output on an oscilloscope and confirming symmetric modulation behavior with respect to the applied voltage.

## MZM static performance

Figure 2 illustrates the experimental setup used for modulator characterization. The platform was tested at a wavelength of 4 μm using a Fabry-Pérot QCL, configured in a custom-built external cavity to ensure single-mode emission with a tunability of approximately 150 nm and an output power of around 20 mW. Details on the external cavity can be found in the Appendix B. Two DFB QCLs operating at 4.3 μm and 4.5 μm were additionally used to broaden the study of the device performance. A half-wave plate placed after the QCL rotates the polarization to enable efficient coupling into the fundamental TE mode of the waveguide. The beam is coupled into a multimode mid-infrared-compatible $InF_3$ lensed fiber and then focused onto the waveguide facet, coupling into the fundamental TE mode of the waveguide. At the output, light is collected by a second $InF_3$ lensed fiber and either directly injected into an optical spectrum analyzer (OSA), power meter, or collimated and focused onto a mid-infrared photodetector using a fiber collimator and a high-numerical-aperture germanium lens.

To evaluate the modulator's performance, several tests were performed. The optical path difference between the two arms is thermally tunable via the integrated TOPS, which was characterised by varying the applied electrical power and monitoring the corresponding change in the MZM output power. A half-wave power ($P_\pi$) of 1.42 W was extracted, as shown in Fig. 3a. The high power requirement stems from the combined effects of a smaller phase shift per unit temperature change at MIR wavelengths, strong thermal dissipation through the sapphire substrate, and the need to keep the thermal electrode sufficiently separated from the waveguide to prevent plasmonic loss of the guided light. To extract the half-wave voltage ($V_\pi$), the MZM was driven with a 100 kHz triangular waveform from an arbitrary function generator and amplified using a high-voltage amplifier. The resulting optical signal was detected by a mercury cadmium telluride (MCT) detector and recorded on an oscilloscope. The measured response is

presented in Fig. 3b. The presented modulator exhibits a $V_\pi L$ of 22.4 V·cm, with an electrode length of 0.8 cm and a gap of 10.5 μm, operating at a wavelength of 4 μm. The measured extinction ratio reaches 17.1 dB, highlighting the fabrication quality of the device. The modulated output power of the device reaches 350 μW, enabling straightforward characterization with a power meter, without the need for a mechanical chopper.

As shown in Fig. 3c, simulations were performed to minimize optical propagation loss while maintaining efficient modulation performance, characterized by the half-wave voltage-length product ($V_\pi L$), for both thin-film thicknesses. The simulated optical mode loss (see top panel of Fig. 3c) increases significantly when the electrode spacing ($G$) is reduced below 12 μm and 8 μm for the 0.9 μm and 1.5 μm films, respectively, due to plasmonic absorption. The thicker 1.5 μm film provides improved optical confinement within the waveguide, whereas optical confinement in the 0.9 μm film is reduced, leading to a larger mode overlap with the sapphire substrate, which does not contribute to the modulation process. This improved confinement allows the electrodes to be placed closer to the waveguide without incurring high optical loss. Specifically, gaps of 13.2 μm and 15.6 μm were used for the 0.9 μm thin film, while gaps between 10 and 12 μm were selected for the 1.5 μm thin film. Bottom panel of Fig. 3c presents both simulated and experimental results for the achieved $V_\pi L$. Simulations show good agreement with the experimental results for both the 0.9 μm and the 1.5 μm devices. With the 1.5 μm film, a minimum of 18.4 V cm was obtained for the smallest-gap modulator, while the 0.9 μm film yielded 31.4 V·cm. The modulator with the lowest $V_\pi$ exhibited moderate output power of approximately 150 μW due to increased plasmonic loss caused by a slight electrode misalignment during fabrication. This justified the choice to present results from the modulator with the highest optical output power instead. As shown by comparison with the simulations, conservative design choices were made to ensure low optical loss, tolerance to fabrication variations, and a clear demonstration of the viability of the proposed device architecture, especially given the moderate efficiency of typical mid-infrared detectors.

As shown on the top panel of Fig. 3d, we demonstrate the operational bandwidth of the MZM from 3.95 μm to 4.5 μm by

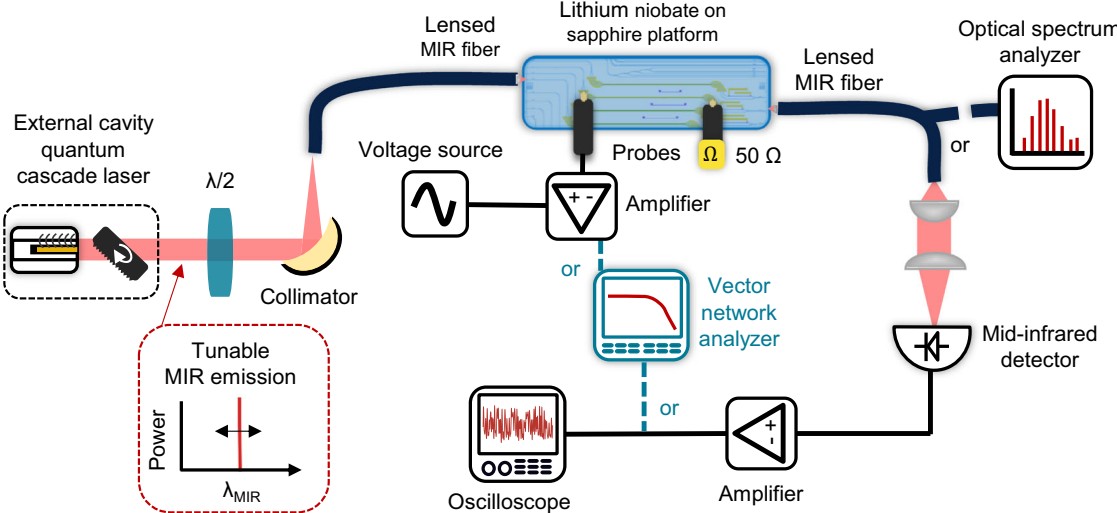

**Fig. 2 | Schematic of the characterization setup for the MIR integrated MZM.** The output of an external cavity laser around 4 µm is coupled into a MIR lensed fiber and then into the photonic chip. At the output, the light is collected using another lensed fiber. The output from the fiber is either directly injected into an optical spectrum analyzer (OSA), a power meter, or collimated and focused onto a high-speed MIR photodetector. This setup can be adapted for both static and dynamic characterization.

evaluating the $V_\pi L$ as a function of wavelength. This corresponds to an impressive 500 nm bandwidth. The bandwidth could potentially extend beyond this wavelength range; however, this region could not be characterized due to the lack of available monomode optical sources at those wavelengths in our laboratory. The slight increase in the value of $V_\pi L$ with wavelength is due to reduced confinement of the optical mode and the lower induced phase shift at longer wavelengths. As shown in the bottom panel of Fig. 3d, the waveguide loss was evaluated at different wavelengths using mid-infrared grating couplers. Additional information on the grating couplers can be found in the Appendix C. The loss is measured to be 1 dB/cm at a wavelength of 3.7 µm, and progressively increases to 5.3 dB/cm at 4.5 µm. The loss at the designed operating wavelength of 4 µm was measured to be roughly 1.1 dB/cm. The losses are hypothesized to be dominated by sidewall roughness[44,45] and could be further reduced through improved fabrication techniques, as theoretical studies suggest that losses approaching 0.1 dB/cm may be achievable[40]. Moreover, adopting a multimode waveguide geometry could further reduce these losses, since the optical mode becomes less sensitive to scattering at the waveguide boundaries, although this would reduce the efficiency of the presented modulator[46]. Finally, we measured a total fiber-to-fiber insertion loss of 14.1 dB at 4 µm, 17.5 dB at 4.3 µm, and around 24.0 dB at 4.5 µm, which includes coupling losses, waveguide propagation losses, and additional plasmonic losses due to the electrodes. The lower loss at 4 µm results from the waveguide being optimized at this wavelength in terms of top width and mode confinement. As the wavelength increases beyond the design point, the mode becomes less confined within the waveguide, leading to higher insertion loss and higher propagation loss. Using the intercept from the cutback measurement with facet coupling, we can remove the coupling contributions and evaluate the intrinsic device loss, including both waveguide and plasmonic electrode losses, to be at most 4 dB at 4 µm and 5 dB at 4.3 µm. The insertion loss from facet coupling has not been evaluated at 4.5 µm, but is expected to be higher. It is important to note that the present design is optimized for operation at a wavelength of 4 µm. The performance at other wavelengths, particularly in terms of insertion and propagation losses, can be improved by optimizing the device geometry for each specific wavelength.

## MZM high speed performance

The high-speed electro-optic performance of the modulators was characterized using $S_{21}$, which represents the small-signal transfer function from the electrical input to the electrical output of a photodetector monitoring the modulated optical signal. Several devices with varying film thickness and electrode lengths of $L = 0.4$ cm, $L = 0.5$ cm, $L = 0.66$ cm, and $L = 0.8$ cm, and gaps were evaluated. The travelling wave electrodes were externally terminated with a 50 Ω RF load to avoid reflection at the end of the electrode. Additional details on the RF design are provided in the Methods and Appendix E. Electro-optic bandwidth measurements were carried out using a Vector Network Analyzer (VNA) in combination with an InAs/InAsSb type-II superlattice (T2SL) photodetector, which exhibits a bandwidth exceeding 20 GHz. The detector demonstrates a high responsivity of over 0.8 A/W at 4 µm when biased above 2 V, as verified through FTIR measurements. Detailed characterization of the detector can be found in the Appendix C. The VNA output was amplified using a 30 dB RF high-power amplifier with a saturation power of approximately 20 dBm. The 0.9 µm platform exhibited a maximum bandwidth of up to 16 GHz, with a small dependence on electrode length, as shown in Fig. 4a. The observed increase in noise around 12.5 GHz was attributed to the onset of antenna-like behaviour, whereby the detector picks up radiation from standing waves present in the RF cables and probes. The restricted bandwidth was attributed to a 50 Ω mismatch for the 0.9 µm configuration. The 1.5 µm platform was subsequently engineered with optimized signal and ground pad geometries to enhance radio-frequency performance. As shown in Fig. 4b, this design yields a nearly flat frequency response up to 20 GHz, which is limited by both the VNA and the detector's intrinsic 3 dB bandwidth. The presented response has been normalized by the detector and amplifier chain, ensuring that the extracted 3 dB bandwidth reflects the modulator's intrinsic performance. No roll-off was observed up to 20 GHz, suggesting that the actual 3 dB bandwidth of the modulator likely exceeds this value. Further characterization at higher frequencies was limited by the absence of a higher bandwidth VNA source and the cutoff frequency of the detector.

To validate the applicability of our modulator, we conducted a free-space transmission experiment over half a meter using a two-level 10 Gbit/s pseudo-random bit sequence of length $2^{15}$, generated by an

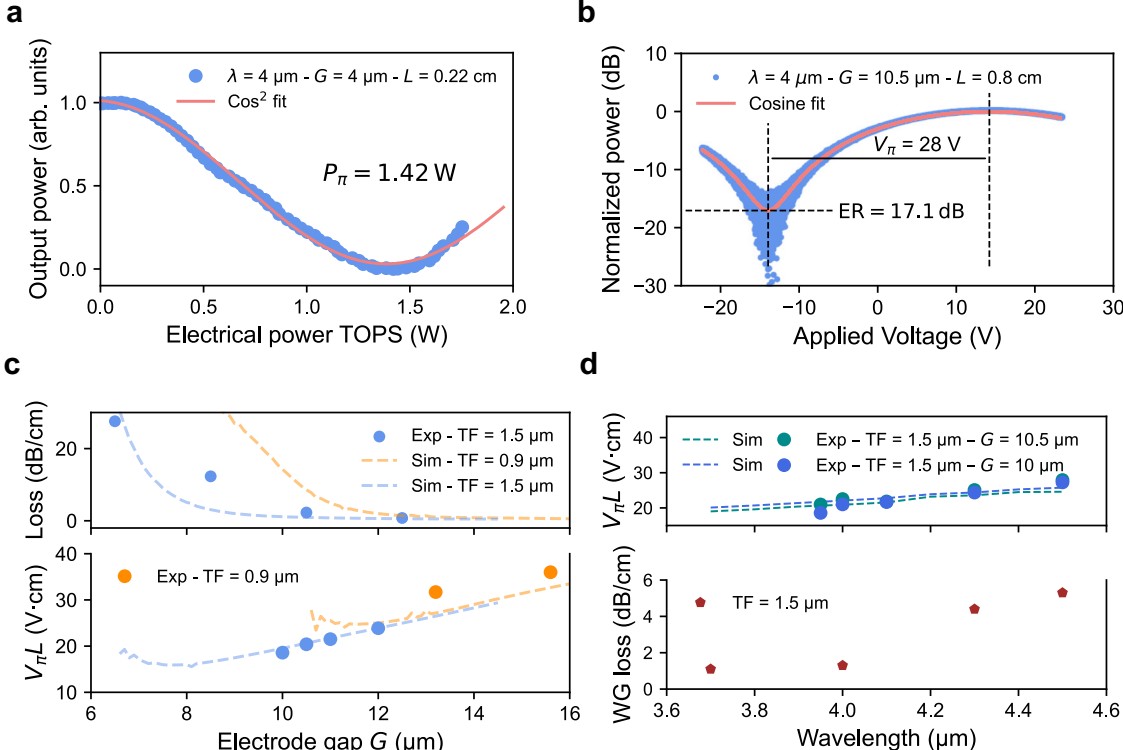

**Fig. 3 | Static electro-optic performance. a** Normalized optical transmission of the MZM as a function of the electrical power applied to the TOPS. The 4 μm transmitted power (blue) was fitted to a cos² function, yielding a $P_\pi$ of 1.43 W for a TOPS with a length of 1.34 mm and an electrode gap of 4 μm. **b** Normalized optical transmission of a modulator with an 10.5 μm electrode gap and thin film thickness of 1.5 μm, exhibiting a $V_\pi$ of 28 V, corresponding to a $V_\pi L$ of 22 V cm, and an extinction ratio of 17 dB. **c** Simulated and experimental results showing the loss variation (top figure) as a function of the electrode spacing, used to optimize the electrode gap G between the signal and ground electrodes, for lithium niobate thin films with thicknesses of 0.9 μm and 1.5 μm, operating at a wavelength of 4 μm. The bottom figure shows both the simulated and experimental $V_\pi L$ products and the simulated propagation loss as functions of the electrode gap for each platform. The experimental $V_\pi L$ values show strong agreement with the simulations, validating the accuracy of the design methodology. **d** The top figure shows the $V\pi L$ evaluation of two different modulators as a function of wavelength, ranging from 3.95 to 4.5 μm. Simulated values for each modulator are represented by dashed lines in the plot. The value increases slightly with wavelength due to reduced confinement of the optical mode. The bottom figure shows the waveguide loss as a function of wavelength from 3.7 to 4.5 μm, measured using a cutback method with mid-infrared grating couplers. The evaluated waveguide loss can be as low as 1 dB/cm at 3.7 μm and gradually increases to around 5 dB/cm at 4.5 μm. Additional information on the grating couplers is provided in the Appendix C.

arbitrary waveform generator (AWG) operating at a sampling rate of 20 GS/s. To evaluate the transmission quality, we present an eye diagram along with the corresponding bit error rate as shown in the Fig. 4c. The total transmitted bit sequence comprised approximately $10^6$ bits. The primary limitation in achieving higher bit rates was the AWG sampling rate of 20 GS/s. Additionally, the implementation of higher-order modulation formats, such as 4-level or 8-level schemes, was constrained by the signal-to-noise ratio of our system.

Using our modulator, we successfully generated an optical frequency comb centered around the carrier frequency for three different modulation frequencies, as measured by the OSA with a resolution of 1.9 GHz. As shown in Fig. 4d, the comb is centered at a wavelength of approximately 4 μm, with symmetric sidebands clearly resolved, confirming the expected modulation behaviour. At 7 GHz, based on the relative powers of the carrier and higher-order sidebands, we observed a maximum attenuation of 6.4 dB per sideband peak, leading to a spectral bandwidth of around 70 GHz, which reflects efficient energy transfer from the central mode into the sidebands. The energy transfer, highlighted by the slope of the comb, decreases with increasing frequency at 10 GHz and 12 GHz, which we attribute to the combined effects of a higher effective $V_\pi L$ at elevated frequencies and reduced electrical power delivery to the electrodes due to impedance mismatch. The measured spectrum is not limited by the OSA noise floor, which is around −70 dBm. In fact, a broader comb could be obtained using a laser source with lower intrinsic amplitude noise. The relatively

high noise level, around -50 dBm in our measurements, arises from the use of the EC-QCL, which typically exhibits higher amplitude and frequency noise due to mechanical tuning elements and longer cavity lengths. For practical spectroscopic applications, a low-noise source such as a 4 μm DFB QCL or an interband cascade laser would significantly enhance the signal-to-noise ratio and allow observation of broader comb spectra, with noise down to −60 dBm and bandwidths exceeding 100 GHz. The ability to tune the spacing between comb lines is a powerful tool for scanning absorption bands. These results demonstrate the modulator's capability to generate rich frequency content through deep amplitude modulation, confirming its potential as a building block for mid-infrared frequency-comb generation and broadband spectroscopic systems.

## Discussion

As demonstrated above, LNOS MZMs developed for the mid-infrared range exhibit strong optical modulation performance, beginning with a broad wavelength tunability demonstrated from 3.95 to 4.3 μm. In addition, the achieved state-of-the-art extinction ratio of 17 dB is highly promising for spectroscopic applications. Furthermore, the optical bandwidth exceeds the current state-of-the-art in the MIR by an order of magnitude. The fabrication process demonstrates tight dimensional control, low sidewall roughness, high pattern fidelity, and reproducibility across devices. This ease of fabrication opens the door to a new generation of compact, efficient modulators or other devices for MIR

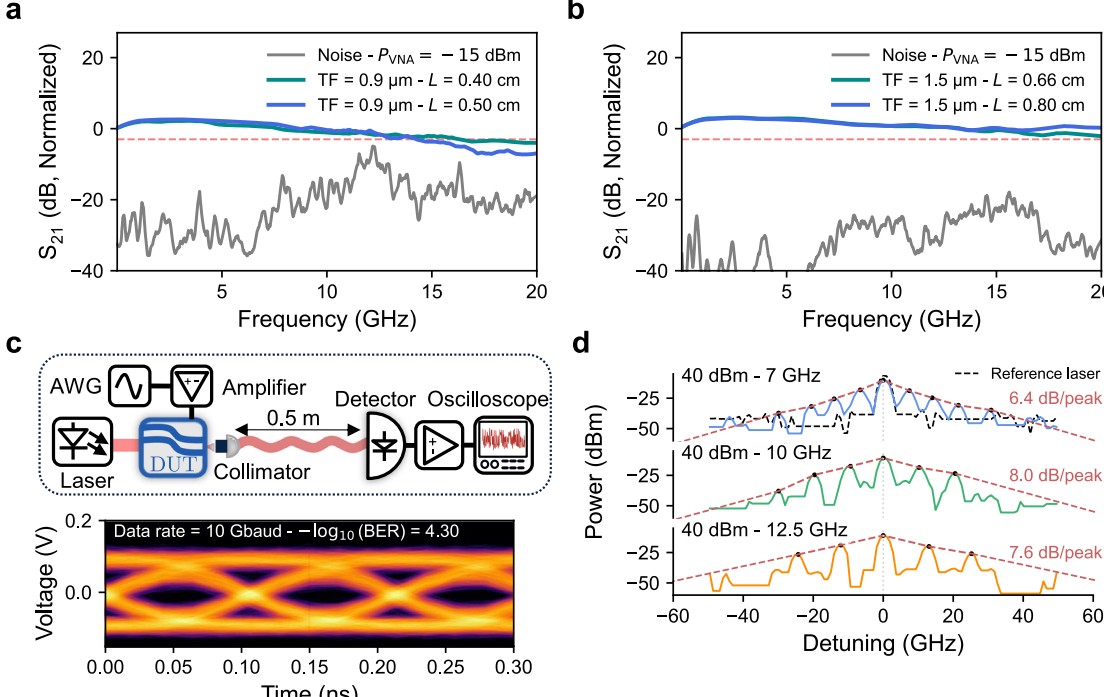

**Fig. 4 | High-speed electro-optic characterization of the MIR MZM. a** Measured $S_{21}$ electro-optic response of two MZMs with different electrode lengths using a 0.9 µm-thick thin film, showing a 3 dB modulation bandwidth up to 15 GHz. The data has been normalized by the detector and amplifier responses to most accurately represent the intrinsic electro-optic bandwidth of the modulators. The restricted bandwidth can be attributed to 50 Ω impedance mismatch. **b** Measured $S_{21}$ electro-optic response of two MIR modulators with different electrode lengths, fabricated on a 1.5 µm thin film. Both devices exhibit a flat modulation response up to 20 GHz. Notably, the reduced $V_\pi$ and lower coupling loss result in a significantly higher signal-to-noise ratio. **c** Top panel: Schematic representation of the free space transmission setup. AWG: Arbitrary Waveform Generator. DUT: Device Under Test.

Bottom panel: 10 Gbit/s free space transmission using the presented modulator at 4 µm exhibiting a clean eye diagram with a bit error rate of $-\log_{10}(\mathrm{BER}) = 4.3$. **d** An optical frequency comb was generated by applying amplitude modulation with a power of 40 dBm at modulation frequencies of 7 GHz, 10 GHz, and 12.5 GHz, centered around 4 µm. The 7 GHz spectrum exhibited eight distinct sidebands with a bandwidth of 70 GHz, indicating efficient modulation-driven energy transfer from the carrier to higher-order sidebands. The black dashed line corresponds to the unmodulated monomode laser spectrum, while the red dashed line illustrates the slope of comb generation. Measurements were performed using an optical spectrum analyzer (OSA) with a spectral resolution of 1.9 GHz.

spectroscopy, sensing, and free-space communication. The observed $V_\pi L$ is higher than that of comparable modulators operating at telecom wavelengths. This behaviour arises from a fundamental limitation the Pockels effect, which induces a change in refractive index $\Delta n$, but the resulting phase shift decreases at longer wavelengths[47,48], as described by:

$$\Delta\phi = \frac{2\pi}{\lambda}\Delta n L \qquad (1)$$

where $\Delta\phi$ is the phase shift, $\lambda$ is the wavelength, and $L$ is the interaction length. Consequently, increasing the wavelength by a factor of approximately 2.5 results in a 3–4 × increase in the intrinsic $V_\pi L$ value[49]. This relationship not only explains the observed performance degradation compared to C-band modulators but also provides a framework for extrapolating the device behaviour beyond the experimentally measured wavelength range. Specifically, it indicates that improved performance can be expected at shorter wavelengths, which could not be experimentally verified due to the absence of a suitable optical source. The measured $V_\pi$ for these devices has been improved by employing a thicker LN film which enhances optical mode confinement, enhances modulation efficiency, and enables a reduction in $V_\pi$, achieving a $V_\pi L$ below 20 V cm. However, this study reveals an intrinsic limitation of $V_\pi$ arising from the operating wavelength and the optical mode confinement. Consequently, future efforts will focus on implementing strategies to significantly lower the required drive voltage, such as resonantly enhanced MZM architectures or

engineered material deposition to improve the electro-optic overlap between the radio-frequency and optical fields. A reduced $V_\pi$ would not only enhance the overall modulation efficiency but also enable new functionalities, including the generation of efficient mid-infrared electro-optic frequency combs in ring resonator configurations. In the context of MIR frequency comb generation, several studies have demonstrated results using QCLs[50], including quantum-walk QCL combs that rely purely on phase modulation[51]. Future improvements in travelling-wave electrode design could enable modulation bandwidths exceeding several tens of gigahertz, making high-speed amplitude modulation feasible. Such a modulator would pave the way for compact generation of mid-infrared pulses with durations on the order of tens of picoseconds, a capability that remains highly challenging today. Current solutions rely on bulky OPOs[52] or external pulse compression of QCL-based frequency combs[53,54]. However, very recent results have shown that racetrack QCLs can generate soliton-based pulses with picosecond-scale durations[55], offering a promising alternative for compact MIR pulse generation. In parallel, our work represents a first step toward the realization of amplitude-modulated MIR frequency combs based on Kerr resonators, leveraging the strong $\chi^{(3)}$ nonlinearity of lithium niobate.

## Methods
### Device fabrication
The chip was fabricated from a commercially available wafer with an x-cut (NGK Insulator, LTD), 900 or 1500-µm LN layer, and a sapphire substrate. Photonic waveguides were defined using a polymer resist

with 100 kV electron beam lithography (Raith EBPG 5200+). They are then dry etched to a depth of 1 μm using Ar+ plasma reactive ion etching (Oxford Instruments PlasmaPro 100). The remaining etch mask and the redeposited material after the etching process were removed using RCA-SCA1 cleaning and buffered oxide etch. Beyond removing redeposition, the RCA-SC1 also reduces sidewall roughness as it slightly etches the LN waveguide itself. After this, the chip is annealed at 500 °C under ambient conditions for 2 hours to heal any damage to the crystal caused by the dry etching process and decrease propagation losses. Thermo-optic phase shifters were then patterned by electron beam lithography to define a resist bilayer, followed by a 100 nm gold evaporation (Evatec 50) and a lift-off process. The electro-optic electrodes are then defined by direct laser writing (DWL 66+), followed by a 900 nm gold evaporation and another lift-off process. Finally, the chip was diced (Disco DAD 3221) to expose the waveguide facets, which were then mechanically polished (Allied Multiprep System 8") using diamond abrasive to minimize in- and out-coupling losses. To further improve facet quality, FIB milling (TFS Helios 5 UX) was performed as a final step.

## Design of the MZM

The performance of passive components, including waveguides and MMIs, was estimated using an eigenmode expansion solver, Ansys Optics. Low-loss and single-mode operation was ensured through careful selection of waveguide width and etch depth. For the 0.9 μm film, the chosen geometry consists of a waveguide top width of $w_{ridge} = 4$ μm and an etch depth of $h_{ridge} = 0.4$ μm, while for the 1.5 μm film, the geometry is defined by $w_{ridge} = 2.5$ μm and $h_{ridge} = 0.92$ μm. The design was chosen to ensure single-mode optical confinement in the fundamental transverse electric mode. Electro-optic simulations using using the software COMSOL Multiphysics and Ansys Lumerical were used to evaluate modulation efficiency and optical field confinement, and optimize electro-optic and thermo-optic performance. Here, a compromise exists when choosing the gap between the waveguides and the electrodes. Reducing the gap increases the overlap between the optical and RF modes (resulting in a lower $V_{\pi}L$). However, decreasing the gap beyond a certain point drastically increases the plasmonic losses, as shown in Fig. 3c (top and bottom panels). Building on the waveguide design optimized for single-mode operation and defined gap spacing, the widths and spacings of the transmission lines were optimized through RF-optical simulations to ensure 50 $\Omega$ impedance matching and velocity matching with the optical mode using COMSOL Multiphysics, thereby reducing reflections and enhancing the modulation bandwidth. Additional details on the electrode design are provided in Appendix E.

## External cavity quantum cascade laser

To enable precise wavelength tuning around 4 μm, an external cavity configuration in the Littrow geometry was implemented. The cavity consists of a QCL coupled to a ruled reflection grating (GR2550-30035) with a dispersion of 2.86 nm/mrad, mounted on a motorized precision rotation stage (PI RS-40). The first-order diffraction from the grating is fed back into the QCL to define the lasing wavelength, while a 50:50 beamsplitter positioned between the laser and the grating is used to extract the output power. Simultaneously, the zeroth-order reflection is used to monitor the injected signal spectrum in real time. The external cavity operates in continuous-wave mode with a tunable bandwidth of approximately 150 nm and an output power exceeding 20 mW, enabling fine control of the emission wavelength via adjustment of the grating angle. The output of the external cavity is then coupled into an InF₃ lensed fiber (Le Verre Fluoré). A schematic of the external cavity and basic characterization can be found in Appendix B. For performance evaluation at 4.0 μm and 4.3 μm, distributed-feedback QCLs with maximum output powers of approximately 40 mW and 60 mW, respectively, were used.

## Static characterization

Static characterization was performed using an arbitrary waveform generator (AFG-2125) in combination with a high-voltage amplifier (Thorlabs HVA200) to generate high-power triangular waveforms. The setup is capable of delivering a maximum electrical voltage of 200 V. The optical output from the external cavity was injected into the system using the same experimental configuration described previously. To characterize the half-wave voltage $V_{\pi}$ and the extinction ratio, an MCT detector (Vigo PVM-2TE-10.6) was used to simultaneously recover the DC and AC electrical signal from the detected modulated optical signal. The signal was recorded using an oscilloscope (TBS 2000B) and subsequently processed with a custom Python script.

## InAs/InAsSb type-II superlattice photodetector

InAs/InAsSb type-II superlattice photodetector were fabricated by the ShanghaiTech Material and Device Lab (SMDL). It was grown by molecular beam epitaxy (MBE) on an n-type GaSb substrate. The layer stack consisted of a 200 nm GaSb buffer, a 300 nm InAs/AlAsSb (1.5/1.5 nm) superlattice bottom contact, a 50 nm n-type InAsSb layer, and a 420 nm unintentionally doped InAsSb drift region. The absorption region comprised a 900 nm InAs/InAsSb (2.9/1 nm) type-II superlattice with a four-step graded doping profile. An AlAsSb/AlSb (1.22/0.61 nm) superlattice served as an electron barrier, capped by a 10 nm InAsSb top contact layer. Device mesas were defined using standard photolithography and inductively coupled plasma dry etching. Ohmic contacts were formed by Ti/Pt/Au deposition. The device surface was then passivated with SU-8, and coplanar waveguide pads were fabricated via air-bridging to provide a 50 $\Omega$ impedance suitable for high-frequency measurements. At room temperature and under a −1 V bias, the device demonstrated a responsivity of approximately 0.6 A/W at 4.5 μm. The cut-off wavelength was found to be around 5.5 μm at room temperature. Temperature-dependent dark current measurements showed that the current density increased from 0.02 A/cm$^2$ at 77 K to 3.94 A/cm$^2$ at 300 K under a −1 V bias. Arrhenius analysis indicated that the dark current is dominated by diffusion processes at high temperatures, with tunneling mechanisms becoming significant below 130 K. For high-speed characterization of the modulator, devices with a diameter of 20 μm were used. To facilitate integration and simplify the experimental setup, the diced device was wire-bonded to a printed circuit board equipped with a standard SMA port, allowing for direct connection to measurement equipment. Both the on-chip and SMA-packaged photodetectors were evaluated using a Lightwave Component Analyzer system, which consisted of a 67 GHz VNA and a 1550 nm laser. The 3 dB bandwidth of the unpackaged (on-chip) device is provided in references[56,57], while the packaged device exhibited a 3-dB bandwidth of 20.3 GHz at a −5 V bias.

## High speed characterization

High-speed characterization was performed using a vector network analyzer (Keysight P5004A) with a maximum bandwidth of 20 GHz. The output port of the VNA was connected to a modulator driver amplifier (AT Microwaves AT-BBLF-0020-3022B) with a bandwidth of approximately 20 GHz and a saturation output power of around 20 dBm. The optical coupling setup remained the same as in the static characterization. The output light is send through a germanium fiber collimator, then focused with a germanium lens with a high numerical aperture (NA = 0.83) and a focal length of 1.873 mm onto the 20 μm$^2$ active area of the detector. The electrical output of the detector was then amplified using a 35 dB low-noise amplifier (AT Microwaves AT-LNA-0043-3504Y) with a 3 dB bandwidth of 43.5 GHz, which was subsequently connected to the input port of the VNA. Both amplifiers and the detector were characterized independently, and their frequency responses were used to normalize the measured $S_{21}$ system response, allowing recovery of the intrinsic bandwidth of the

modulator. For the transmission experiment, a 20 GS/s arbitrary waveform generator (ARB Rider AWG-7204) was used in conjunction with a 30 GHz oscilloscope (Tektronix DPO77002SX). For frequency comb generation, the signal from a signal generator (Rohde & Schwarz SGS100A) was amplified using a high-power amplifier (Fairview Microwave FMAM513), and the output was measured with a Thorlabs OSA 305 operating at a resolution of 1.9 GHz.

## Data availability

The data supporting the plots within this article are available from the corresponding author upon request.

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

## Acknowledgements

The MWIR photodiode chip was fabricated with support from the ShanghaiTech Material and Device Lab (SMDL). The authors thank Dimitrios Granouzis Solounias for developing the Python program used to control the external cavity quantum cascade laser. We gratefully acknowledge support for the fabrication and characterization of our samples from the Scientific Center for Optical and Electron Microscopy (ScopeM), the BRNC and FIRST cleanroom facilities at ETH Zurich, and IBM Rüschlikon. We acknowledge support from the Schweizerischer Nationalfonds (SNF) through a postdoctoral fellowship under the HARMONY project. We also acknowledge the National Key Laboratory of Infrared Detection Technologies (IRDT-ZGKXY-25-12). We thank Dr. Antonis Olziersky for support with the electron beam lithography process.

## Author contributions

P.D., P.J., and O.P. conceived the project. P.D., O.P., and T.K. carried out simulations and designed the TFLN modulator. P.J. fabricated the modulator. P.D. and P.J. conducted the experiments. P.D. and P.J. analyzed and discussed the experimental results. M.Bertrand. and J.K. assisted with the experiments. M.Bertrand. provided the mid infrared QCL, which was grown by M.Beck. Z.D. and B.C. provided the high speed photodiode. P.D. and P.J. wrote the manuscript. M.Bertrand., O.P., J.K., J.F., and R.G. reviewed the manuscript. The project was carried out under the supervision of R.G.

## Funding

## Competing interests

The authors declare no competing interests.
