## [Transparent Peer Review file · Nature Communications]

Thin film lithium niobate on sapphire for integrated mid-infrared modulator

Corresponding Author: Dr Pierre Didier

A version of this paper was originally rejected for publication by Nature Communications, however that decision was reconsidered after appeal by the authors.

Version 1:

Reviewer comments:

Reviewer #1

(Remarks to the Author)

The authors report a high-performance MIR MZM based on an LNOS platform. For this work to be considered for publication in a high-impact journal like Nature Communications, the following issues require careful attention and substantial revision.

1. First, could the authors clarify the specific requirements for high-speed electro-optic modulators in application scenarios such as molecular spectroscopy detection, free-space optical communication, and MIR spectroscopy? For these applications, is high modulation efficiency more practical than high modulation speed?
2. The authors should provide a more balanced and comprehensive evaluation of prior art. While it is reasonable to highlight the limitations of previous works to contextualize their own advances, a fair discussion should also acknowledge the pioneering role and significant contributions of those earlier studies to the field.
3. The use of "half-wave voltage of 22 V·cm" in the abstract is incorrect, as the unit "V·cm" characterizes modulation efficiency, not the half-wave voltage itself. This same conceptual confusion appears on page 3, where " $V\pi$ L below 1 V" is stated. The product $V\pi \cdot L$ is a metric for efficiency, and its value cannot be directly compared to a voltage (V). These should be corrected to avoid misunderstanding.
4. The abstract states: "Furthermore, we demonstrate full π -phase modulation in the MIR, representing a key milestone for integrated MIR photonics." Figure 3b shows that a π -phase shift is achieved within a bias voltage range of -14 V to 14 V. However, in Reference 19, a 0.75 π phase shift is achieved with a bias range of 0 V to 3.5 V, and in Reference 20, a π -phase shift is achieved with a bias range of 0 V to 1.2 V. Should the claim of achieving "full π -phase modulation in the MIR as a key milestone" be qualified with more specific context or limitations?
5. The captions for Fig. 4c and 4b appear to be swapped.
6. Equation 1 in the Discussion section is overly fundamental. Is it necessary to include it separately in this context?
7. The caption for Fig. B2, "Fabrication process of TFLN on sapphire," does not match the content of the figure.
8. Could the authors attempt to subtract the S21 curve of the detector from the current S21 curve to further quantify the modulator's bandwidth?
9. To significantly enhance the impact of this work, it is highly recommended that the authors include a demonstration of their high-speed MIR MZM in a practical application. Given the potential applications mentioned in the text, implementing a proof-of-concept experiment in areas such as MIR spectroscopy, gas sensing, or free-space data transmission would compellingly validate the device's utility and performance.

Reviewer #2

(Remarks to the Author)

Reviewer Report

Original and Timely: 3 – Probably

Lasting Impact: No

Convincing and Rigorous Data: Yes, but requires some revision

Interest to the Community: Very Likely

Sufficient Information: No

Satisfactory English: Yes

Appropriate Title: Yes

Good Abstract: Yes

Clear Figures: Yes

Adequate References: No

This manuscript presents an integrated electro-optic modulator on a TFLN-on-sapphire (LNOS) platform, demonstrating low loss and high bandwidth operation in the mid-infrared (MIR) regime. Since MIR photonics remains a challenging field due to the lack of mature components such as lasers, detectors, and fibers, this work is technically well-executed and of potential interest to the community.

However, similar integrated electro-optic modulators have already been reported. Therefore, to meet the publication standards of Nature Communications, the manuscript would benefit from additional experimental data and further discussion to clearly establish novelty and impact.

Below are detailed comments:

1. Title clarity:

Since a mid-IR modulator based on TFLN has already been demonstrated in the following paper, the title should be more specific, explicitly indicating that the work is based on the LNOS platform:

Hyeon Hwang, Kiyong Ko, et al., "A wide-spectrum mid-infrared electro-optic intensity modulator employing a two-point coupled lithium niobate racetrack resonator," *APL Photonics* 10, 016116 (2025).

2. Overstated claim:

The sentence on page 4 —

"To our knowledge, this represents the first photonic integrated MZM demonstrated in the MIR, surpassing previous demonstrated modulators in terms of speed and extinction ratio by nearly two orders of magnitude."

— should be removed, as it is not accurate in light of prior demonstrations.

3. Insufficient references for MIR relevance:

The following statement is correct but lacks supporting citations:

"In this context, phase and amplitude modulation are essential techniques for MIR spectroscopy – improving sensitivity, resolution, and signal-to-noise ratio."

Please include appropriate references that emphasize the importance of MIR modulation and its applications.

4. Spectral limitation and platform comparison:

The experiments were conducted only between 3.9–4.1 μm , covering a narrow range with limited molecular fingerprint information. To better demonstrate the potential broadband applicability of the platform, I suggest adding theoretical loss calculations for both LNOI and LNOS platforms across the 3–5 μm range (e.g., in Fig. 1). Additionally, including Q-factor measurements from ring resonators would help substantiate the claim of achieving a low-loss MIR platform.

5. Inconsistency in extinction ratio data:

In Fig. 3b, the extinction ratio (ER) is labeled as 34.1 dB, but the data suggest a minimum transmission of ~ 0.1 (corresponding to ~ 10 dB ER). The same issue appears in Fig. E5. Please verify and clarify this discrepancy. It may also be more appropriate to plot the modulation data on a logarithmic (dB) scale to better visualize the claimed high ER performance.

6. Missing references for wavelength-dependent modulation:

The following statement on page 9 should be supported by experimental references:

"This behaviour arises from a fundamental limitation — the Pockels effect induces a change in refractive index Δn , but the resulting phase shift decreases at longer wavelengths, as described by:"

I recommend citing the following works:

o Hixin Xue et al., "Full-spectrum visible electro-optic modulator," *Optica* 10, 125–126 (2023).

o Hyeon Hwang, Kiyong Ko, et al., "A wide-spectrum mid-infrared electro-optic intensity modulator employing a two-point coupled lithium niobate racetrack resonator," *APL Photonics* 10, 016116 (2025).

7. Discussion of the main limitation (high $V\pi$):

As discussed in the manuscript, the major limitation of the current platform is its high half-wave voltage ($V\pi$). It would be helpful to include additional information in the Supplementary Material regarding possible improvement limits through electrode design optimization. From a practical standpoint, it appears that the achievable enhancement may already be constrained by the inherent characteristics of MIR wavelengths.

Overall evaluation:

This is a well-executed and potentially valuable study demonstrating MIR modulation on an LNOS platform. However, in its current form, the manuscript lacks sufficient distinction from prior work and requires additional data, clarifications, and references to convincingly justify publication in Nature Communications.

Reviewer #3

(Remarks to the Author)

In the manuscript "Integrated thin film lithium niobate mid-infrared modulator", the authors report a Mach-Zehnder electro-optic modulator (MZM) on lithium niobate on sapphire (LNOS) nanophotonic platform, using which they demonstrate phase and amplitude modulation in the mid-infrared (MIR), around 4-microns wavelength. They further demonstrate data-transmission and generation of an electro-optic (EO) comb in this spectral region.

Upon reading this manuscript, I have the following questions and feedback for the authors:

- Would it be possible for the authors to discuss in more detail the design considerations of their device, perhaps in section

2.1, or in a separate section in the appendix? I believe it would be helpful for the reader to understand how key figures of merit, such as the $V_{\pi L}$ and EO bandwidth, are dependent on the various design parameters, why particular waveguide dimensions and electrode dimensions were chosen, and if special attention was paid to group velocity matching and impedance matching. While some of these design considerations and simulations are interspersed with results, I think it'd be beneficial to have most of them collected in one place to provide a full picture of designing an MZM in the MIR, and to understand if any design optima may differ from those that are well-established in the near infrared.

- The authors use a MIR-compatible multimode lensed fiber to couple light into the input waveguide. How do they ensure that they only have fundamental mode excitation in the waveguide?
- Fig 3d has only four wavelengths characterized between 3.95 and 4.3 microns. Based on the QCL spectrum in appendix B, would it be possible to have measurements at more wavelengths in this range? It would be good to understand the apparent dispersive trends and ascertain if they arise from experimental noise or if they correspond to oscillations in the device efficiency.
- The generation of MIR EO comb through the MZM is certainly of great interest due to its potential applications, however this portion of the manuscript appears to be lacking in some details, such as the driving voltage and observed bandwidth, as well as in discussions about ways to improve the bandwidth of the comb. Also, can multiple devices (varying waveguide thickness, electrode length, etc.) be tested for EO comb generation and the results compared?
- From the transcript: "While the simulation shows good agreement with the experimental results for the 1.5 μm devices, a slight discrepancy is observed for the 0.9 μm case, likely due to a non-uniform gap length along the electrode."; why would the gap between lithographically defined electrodes be nonuniform? Could it be due to improper metal liftoff? An optical microscope observation might provide some clarification.
- Based on the propagation loss estimates from the cutback measurements, can the authors ascertain the excess loss through the MZM and if / how it depends on the waveguide and electrode designs?

Some minor comments:

- It appears from Fig. 1b that the waveguide width is 4 microns, while I believe the intention there is to show the wavelength?
- The layout of the electro-optic electrode is not clear to me from figure 1d; perhaps the authors could change the color / contrast of the image to make the photonics components more visible in this image?
- Typo in figure 3b caption, V_{π} of 28 V (not Vcm)?
- Figure 3c caption needs some clarification. I would also recommend that the authors plot the simulations as solid / dashed lines and reserve cross / dot for experimental data points.
- Captions for Fig.4c&d appear to be interchanged.
- Fig. B2 caption incorrect.
- Please clarify the TOPS set point for the EO measurements.

Overall, my impression is that while the authors have adequately demonstrated MZM in the LNOS platform in the MIR, if they could expand upon their demonstration and / or analysis of MIR EO-comb generation in this platform, it would substantially increase the contribution of their work and make it more suitable for publication in Nature Communications.

Version 2:

Reviewer comments:

Reviewer #1

(Remarks to the Author)

The authors have adequately addressed my previous comments. Regarding the revised manuscript, I would like to suggest one minor improvement. To better illustrate the experimental configuration for the free-space transmission at 10 Gbit/s over a 0.5 m distance, the addition of a figure showing a photograph or a link diagram of the actual setup would be valuable. Such a depiction would greatly aid in the interpretation and visualization of this specific experimental scenario.

Reviewer #2

(Remarks to the Author)

I appreciate the authors' detailed response and the revisions made to the manuscript. I understand the authors' position that a full demonstration of spectroscopy applications is beyond the scope of this work and that the manuscript aims to serve as a "foundational demonstration" of the LNOS platform in the mid-infrared.

However, if the focus is shifted towards the fundamental aspects of the platform rather than practical applications, the analysis of the optical loss mechanisms must be significantly more rigorous.

Specifically, I strongly suggest adding a fundamental analysis comparing the experimental losses with the theoretical limits. For instance, based on Mishra, Jatadhari, et al., *Optica* 8.6 (2021): 921-924, the calculated absorption loss of the LNOS platform is approximately ~ 0.01 dB/cm at 3.5 μm and ~ 0.1 dB/cm at 4 μm .

In contrast, the propagation losses reported in this manuscript (e.g., ~ 1.3 dB/cm at 4 μm) are significantly higher than these theoretical limits. This discrepancy suggests that the current device does not fully exploit the intrinsic low-loss advantage of the LNOS platform. In fact, such loss values might be comparable to what could be achieved on LNOI platforms in certain conditions, thereby weakening the argument for LNOS.

Therefore, to justify the significance of this work as a "foundational demonstration," the authors must:

1. Explicitly compare their measured losses with the theoretical absorption limits calculated in literature (e.g., Mishra et al.).
2. Provide a detailed analysis of the sources of the excess loss (e.g., scattering due to sidewall roughness vs. material absorption).
3. Discuss why the current experimental values deviate from the theoretical potential and how this gap can be bridged in future work.

Reviewer #3

(Remarks to the Author)

The authors have adequately addressed my queries and improved the manuscript.

Version 3:

Reviewer comments:

Reviewer #2

(Remarks to the Author)

It is acknowledged that the disparity between experimental measurements and calculated values for propagation loss arises from uncontrolled losses introduced during the wafer-level fabrication process. However, the exact role of scattering loss in this discrepancy requires precise formulation. The underlying physics dictates that Rayleigh scattering loss, which is characteristic of surface roughness, scales inversely with the fourth power of the wavelength (λ^{-4}).

By referencing the three cited papers, one can accurately predict the magnitude of scattering loss or, alternatively, derive the wavelength-dependent scaling of this loss mechanism. The paper, "Estimation of Propagation Losses for Narrow Strip and Rib Waveguides," is specifically recommended as a simplified, practical tool for loss estimation.

Crucially, conclusive empirical evidence is required to definitively establish that scattering loss is the singular or dominant root cause of the observed performance limitations.

1. "Roughness-Limited Performance in Ultra-Low-Loss Lithium Niobate Cavities" by Khalatpour et al.
2. "A theoretical analysis of scattering loss from planar optical waveguides" by Payne and Lacey
3. "Estimation of Propagation Losses for Narrow Strip and Rib Waveguides" by Lindecrantz and Hellesø

Response to the Reviewers' comments on the manuscript titled "Integrated thin film lithium niobate mid-infrared modulator" submitted to Nature Communications

Thank you very much for the opportunity to revise our manuscript, "Integrated thin film lithium niobate mid-infrared modulator."

We would like to express our sincere gratitude to the Editor and Reviewers for their thorough evaluation of our manuscript. Below, we provide a detailed, point-by-point response to all the valuable suggestions and questions raised by the Reviewers. Along with this resubmission, we have included a revised version of the manuscript that incorporates these constructive comments. We believe that the revisions and clarifications have substantially improved the quality of our work, bringing it to the high standards of Nature Communications.

In particular, we have:

- Corrected the points on the reference highlighted by the reviewers and improved the state-of-the-art discussion to better contextualize our work. **(Part Main)**
- Extended the experimental modulation measurements up to 4.5 μm , supported by new simulations illustrating the tunability of our device across the mid-infrared range. **(Part 2.2 – Fig 3.d)**
- Employed a mid-infrared grating coupler to precisely measure the waveguide propagation loss, demonstrating state-of-the-art loss of around 1.3 dB/cm at 4 μm and reasonable loss at 4.5 μm (5.2 dB/cm). Included an experimental evaluation of the additional loss introduced by the plasmonic losses caused by the electrodes. **(Part 2.2 – Fig 3.c-d / Fig C3.b-d)**
- Extended the experimental demonstration of comb generation by providing additional details and new measurements that expand the accessible frequency spacing. **(Part 2.3 – Fig 4.d)**
- Included simulations and S11 measurements to validate the RF design methodology of the devices. **(Part Appendix E – Fig E5)**

We also acknowledge the reviewers' request for a more advanced demonstration of frequency comb generation. This is indeed an exciting direction we are actively developing in our laboratory. However, achieving a high-performance comb source requires a different device design and characterization framework than the Mach-Zehnder modulator presented in this work and goes beyond the scope of this work. We therefore view the present manuscript as a foundational demonstration of a low-loss mid-infrared platform, showcasing its efficient Pockels effect and evaluating its low losses in the MIR and related challenges. Demonstrating the feasibility and reliability of a fabrication platform that extends lithium niobate, a leading material in integrated

photonics, into the mid-infrared for nonlinear optical applications (SPDC, DFG process...).

We believe the revised manuscript now substantially strengthens the work and clarifies its significance as a platform demonstration, while laying the groundwork for future developments such as comb generation.

Reviewer #1 (Remarks to the Author):

The authors report a high-performance MIR MZM based on an LNOS platform. For this work to be considered for publication in a high-impact journal like Nature Communications, the following issues require careful attention and substantial revision.

1. First, could the authors clarify the specific requirements for high-speed electro-optic modulators in application scenarios such as molecular spectroscopy detection, free-space optical communication, and MIR spectroscopy? For these applications, is high modulation efficiency more practical than high modulation speed?

Response: The specific requirements for mid-infrared electro-optic modulators indeed depend strongly on the targeted application.

For free-space optical communication, high-speed modulation is essential to achieve high data rates while maintaining sufficient optical output power and extinction ratio to ensure signal integrity and link robustness (Trichili et al., J. Opt. Soc. Am. B 37, A184–A201, 2020). In this context, the high-speed capability and low optical loss demonstrated by our device are directly relevant, as they enable gigabit-per-second MIR data transmission, as shown in our free-space link experiment.

In mid-infrared spectroscopy, many environmentally relevant molecules exhibit strong absorption features in the MIR, up to an order of magnitude stronger than in the near-infrared, enabling more precise and sensitive detection (Haas et al., Annu. Rev. Anal. Chem. 9, 45–68, 2016). A system with a high extinction ratio allows clear discrimination between the measured signal and the noise floor under lock-in detection. Furthermore, high-speed modulation enables the observation of fast transient phenomena (Schultz et al., Spectrochim. Acta A 188, 666–674, 2018) and helps protect samples from damage while mitigating thermal effects (Tamamitsu et al., Nat. Photonics 18, 738–743, 2024).

Phase modulation, on the other hand, enables wavelength modulation spectroscopy (Schilt et al., Appl. Opt. 42, 6728–6738, 2003) and the generation of frequency combs, which are powerful tools for enhancing both the speed and sensitivity of spectroscopic measurements (Picqué & Hänsch, Nat. Photonics 13, 146–157, 2019). In this context, modulation efficiency and speed are crucial, as they determine the achievable

conversion efficiency into comb lines, the line spacing, and consequently the comb bandwidth.

Thus, the present demonstration of a high-speed, low-loss MIR modulator establishes the core capability of the lithium niobate on silicon (LNOS) platform. It provides a foundation upon which optimized designs can be tailored toward either high-speed communication or highly efficient spectroscopic applications, depending on the targeted use case.

These points have been clarified in the revised manuscript to better highlight the importance of a highly efficient, high-speed modulator for mid-infrared applications.

Change in the manuscript:

- *“Furthermore, many environmentally relevant molecules exhibit strong absorption features in the MIR that are up to an order of magnitude stronger than in the near-infrared, enabling more precise and sensitive detection [6, 7]” Part Main*
- *“In this context, efficient electro-optic modulators play a crucial role in both domains. External modulators, in particular, enable high modulation depths and can handle significantly higher optical power than directly modulated sources, making them highly advantageous for achieving strong, high-quality modulated signals in the mid-infrared.[8] The MIR spectral range also offers several benefits for spectroscopy, whereby precise control over the phase and amplitude of mid-infrared light is required. Amplitude modulation enables lock-in detection to improve sensitivity and suppress background signals [9, 10], and it can also be used to probe very fast phenomenon [11] and delicate samples [12]. For example, Tamamitsu et al. use a 0.1 ns (10 GHz) MIR pulse to confine heat diffusion and enhance photothermal efficiency, although it is currently generated by a bulky OPO system. Phase modulation, on the other hand, enables wavelength modulation spectroscopy [12] and the generation of frequency-comb light sources[13], providing powerful tools for high-resolution and broadband spectroscopy” – Part Main*

2.The authors should provide a more balanced and comprehensive evaluation of prior art. While it is reasonable to highlight the limitations of previous works to contextualize their own advances, a fair discussion should also acknowledge the pioneering role and significant contributions of those earlier studies to the field.

Response: We agree with your observation and **have implemented a more balanced discussion that acknowledges the pioneering work in mid-infrared modulator technology.**

3.The use of "half-wave voltage of 22 V-cm" in the abstract is incorrect, as the unit "V-cm" characterizes modulation efficiency, not the half-wave voltage itself. This same conceptual confusion appears on page 3, where " $V\pi$ L below 1 V" is stated. The product $V\pi\cdot L$ is a metric for efficiency, and its value cannot be directly compared to a voltage (V). These should be corrected to avoid misunderstanding.

Response: We agree that $V\pi\cdot L$ represents the half-wave voltage–length product, which is a metric of modulation efficiency rather than the half-wave voltage itself. In our manuscript, we intended to refer to the half-wave voltage–length product rather than only the half-wave voltage. **The text has been revised accordingly to avoid confusion.**

4.The abstract states: "Furthermore, we demonstrate full π -phase modulation in the MIR, representing a key milestone for integrated MIR photonics." Figure 3b shows that a π -phase shift is achieved within a bias voltage range of -14 V to 14 V. However, in Reference 19, a 0.75π phase shift is achieved with a bias range of 0 V to 3.5 V, and in Reference 20, a π -phase shift is achieved with a bias range of 0 V to 1.2 V. Should the claim of achieving "full π -phase modulation in the MIR as a key milestone" be qualified with more specific context or limitations?

Response: We agree that the statement in the abstract should be better qualified to clarify the specific context and significance of our result. In the revised manuscript, we have rephrased this sentence to emphasize that, to the best of our knowledge, no other integrated device operating in the mid-infrared has simultaneously achieved full π -phase modulation, low propagation loss, and high-speed performance.

While References 19 and 20 report lower $V\pi$ values, these devices are based on free-carrier electro-absorption mechanisms rather than the Pockels effect, and therefore suffer from strong free-carrier absorption, which leads to substantial optical loss. For instance, in Ref. 19, a 0.75π phase shift is achieved at 3.5 V, but the modulator exhibits approximately 3.85 dB loss excluding MMI and waveguide losses due to free-carrier absorption alone, and an extinction ratio of 13 dB. Similarly, in Ref. 20, although a π -phase shift is reached with a low drive voltage (1.2 V), the reported excess loss exceeds 22 dB, mainly caused by carrier-induced absorption in the waveguide.

Moreover, the reported waveguide propagation losses in Ref. 20 are 11.5 dB/cm at 3.72 μm and 15.0 dB/cm at 3.88 μm , increasing with wavelength largely due to high material absorption by the SiO₂ buried oxide layer.

In contrast, our work demonstrates a Pockels-based integrated LiNbO₃ modulator with low propagation loss, high speed and broad MIR operation up to 4.5 μm , achieving full π -phase modulation without introducing significant excess absorption. This combination of integration, efficiency, and low loss, all in the mid-infrared regime, constitutes the key milestone highlighted in our manuscript.

The statement has been revised to clarify that, to the best of our knowledge, no integrated device currently demonstrates simultaneously the same combination of performance metrics achieved in our work. While previous reports have shown comparable phase shifts, they either lack full integration, operate at lower speeds, or do not meet all key metrics concurrently. **We have also revised the manuscript to mention in more detail the current state-of-the-art in MIR modulators.**

Changes in the manuscript:

- *“Furthermore, we demonstrate full π -phase modulation in the MIR, uniquely combining integration, low propagation loss, extinction ratio and high-speed operation, representing a key milestone for MIR photonics. These results establish a pathway toward energy-efficient, high-performance mid-infrared photonic systems for applications in telecommunications, sensing, and quantum technologies.” Part Abstract*
- *“On the other hand, phase modulation in the mid-infrared has so far been demonstrated in only a few platforms, including optical parametric oscillators (OPOs) based on periodically poled bulk lithium niobate [31, 32], Stark-effect devices using intersubband transitions [33, 34], and graphene-based structures [35]. While OPOs provide high optical power, they rely on bulky tabletop components and have very high- power consumption. As already discussed, carrier depletion or injection modulators can achieve performance approaching a π -phase shift with lower $V\pi L$ but suffer from extinction ratios around 15 dB and high optical losses [24, 25]” Part Main*
- *“To our knowledge, these values of loss, speed, and extinction ratio have not previously been achieved in a single integrated MZM operating in the MIR [20], albeit with a higher $V\pi L$.” Part Main*

5.The captions for Fig. 4c and 4b appear to be swapped.

Response: We have corrected the caption in the manuscript (Fig. 4c, Fig. 4b)

6.Equation 1 in the Discussion section is overly fundamental. Is it necessary to include it separately in this context?

Response: We agree that Equation 1 presents a fundamental relationship; however, we believe it remains relevant in this context. **In the revised manuscript, we have clarified its purpose by emphasizing that it not only explains the performance degradation compared to similar Pockels-based telecom C-band modulators but also provides a basis for extrapolating the device behavior outside the measured wavelength range.** Specifically, the equation helps predict the decrease of performance for higher wavelength and the improved performance can be expected at shorter wavelengths, which could not be experimentally verified due to the lack of an appropriate optical source.

Changes in the manuscript:

- *“Consequently, increasing the wavelength by a factor of approximately 2.5 results in a 3–4× increase in the intrinsic $V\pi L$ value.[24] This relationship not only explains the observed performance degradation compared to C-band modulators but also provides a framework for extrapolating the device behavior beyond the experimentally measured wavelength range. Specifically, it indicates that improved performance can be expected at shorter wavelengths, which could not be experimentally verified due to the absence of a suitable optical source.” Part Discussion*

7.The caption for Fig. B2, "Fabrication process of TFLN on sapphire," does not match the content of the figure.

Response: We have corrected the caption in the manuscript.

8.Could the authors attempt to subtract the S21 curve of the detector from the current S21 curve to further quantify the modulator's bandwidth?

Response: The detector's S21 response has already been subtracted, and the presented curve therefore represents the modulator's corrected bandwidth. The detector S21 corresponds to its intrinsic response, after removing the contributions from the amplifier and modulator. **The requested clarifications have been incorporated into the revised manuscript.**

Changes in the manuscript:

- *“The data has been normalized by the detector and amplifier responses to most accurately represent the intrinsic electro-optic bandwidth of the modulators.”
Caption of Fig.4*
- *“The presented response has been normalized by the detector and amplifier chain, ensuring that the extracted 3 dB bandwidth reflects the modulator's intrinsic performance.” Part 2.3*

9.To significantly enhance the impact of this work, it is highly recommended that the authors include a demonstration of their high-speed MIR MZM in a practical application. Given the potential applications mentioned in the text, implementing a proof-of-concept experiment in areas such as MIR spectroscopy, gas sensing, or free-space data transmission would compellingly validate the device's utility and performance.

Response: We fully agree that demonstrating a practical application would further highlight the potential of our high-speed MIR Mach–Zehnder modulator. In our current work, we have already demonstrated free-space transmission at 10 Gbit/s (0.5 m distance), which provides first proof of the device's capability for mid-infrared data links (Fig. 4.c). Furthermore, we demonstrate effective comb generation for potential spectroscopy applications. (Fig. 4.d)

Implementing a complete proof-of-concept system for applications such as mid-infrared spectroscopy, gas sensing, or long-distance free-space communication would require specialized setups and instrumentation beyond the scope of the present study. Such demonstrations would also demand additional expertise and complex experimental infrastructures, and are therefore better suited for dedicated follow-up work.

We believe that the current results already establish the innovative low loss platform, high-speed performance and practical potential of our device, laying a solid foundation for future studies which are more application-oriented.

We thank the reviewer for their valuable feedback. In response, we have incorporated additional data and clarifications that directly address the raised concerns and further strengthen the manuscript.

Reviewer #2 (Remarks to the Author):

Reviewer Report

Original and Timely: 3 – Probably

Lasting Impact: No

Convincing and Rigorous Data: Yes, but requires some revision

Interest to the Community: Very Likely

Sufficient Information: No

Satisfactory English: Yes

Appropriate Title: Yes

Good Abstract: Yes

Clear Figures: Yes

Adequate References: No

This manuscript presents an integrated electro-optic modulator on a TFLN-on-sapphire (LNOS) platform, demonstrating low loss and high bandwidth operation in the mid-infrared (MIR) regime. Since MIR photonics remains a challenging field due to the lack of mature components such as lasers, detectors, and fibers, this work is technically well-executed and of potential interest to the community.

However, similar integrated electro-optic modulators have already been reported. Therefore, to meet the publication standards of Nature Communications, the manuscript would benefit from additional experimental data and further discussion to clearly establish novelty and impact.

Response: We apologize if the comparison was not sufficiently clear in the previous version. We are confident that our device marks a key distinction from previously reported technologies. Regarding the references, it is important to distinguish between three categories of modulators:

- LNOI-based modulators are intrinsically limited to operation below approximately 3.8 μm due to strong absorption in the buried silicon dioxide layer. For example, in the very interesting work by Hwang et al. (“A wide-spectrum mid-infrared electro-optic intensity modulator employing a two-point coupled lithium niobate racetrack resonator,” *APL Photonics* 10, 01501 (2025)), the device is based on the lithium-niobate-on-insulator (LNOI) platform, which benefits from mature fabrication processes but suffers from significant absorption in the buried SiO_2 layer in the mid-infrared, as illustrated in the following graph. As shown by these results, the propagation loss already reaches about 4 dB/cm at 3.8 μm , which severely limits operation beyond this wavelength. And the absorption would increase significantly after 3.8 μm as shown in the graph below.

[editorial note: third party material redacted]

Mishra, Jatadhari, et al. *Optica* 8.6 (2021): 921-924

Moreover, while a 100 GHz modulation bandwidth was predicted, only a 1 kHz response was experimentally demonstrated. In contrast, our modulator is implemented on the lithium-niobate-on-sapphire platform, which eliminates the SiO_2 absorption issue (see figure above) and enables low-loss operation well beyond 3.8 μm . Also, achieving high-quality fabrication on the LNOS platform is particularly challenging due to the optical transparency and insulating nature of the sapphire substrate, which complicates alignment during lithography and charge dissipation during processing. An additional fabrication challenge is the deeper lithium niobate etching (>400 nm) required compared to the typical 200–400 nm used in most LNOI devices, to accommodate the longer operating wavelength of our MIR device. Nonetheless, the successful realization of a high-

speed MIR modulator on this platform demonstrates its strong potential for extending electro-optic modulation further into the mid-infrared, while also benefiting from its exceptional nonlinear properties.

- Other lithium niobate technology: Si/LN hybrid platform based on transfer printing, as demonstrated by Xu et al. (Advanced Optical Materials, 2023). This method enables mid-infrared operation up to 3.8 μm and probably higher, achieving a modulation efficiency of 12.3 V-cm, an extinction ratio around 25 dB, and a modulation bandwidth above 5 MHz. While this technique shows promising performance, it involves relatively high fabrication complexity associated with the heterogeneous integration process. Moreover, the reported devices exhibit approximately 3.5 dB/cm higher propagation loss compared to our platform, which may limit overall system efficiency and scalability.
- III-V-based carrier-injection modulators suffer from very high optical loss during modulation because of free-carrier absorption. These devices have demonstrated modulation bandwidths roughly two orders of magnitude lower than those achievable through electro-optic modulation. (Li, Tiantian, et al. "Ge-on-Si modulators operating at mid-infrared wavelengths up to 8 μm ." Photonics Research 7.8 (2019): 828-836.),

Our work represents the first electro-optic Mach-Zehnder modulator enabling efficient (extinction ratio), very high speed (bandwidth), and low-loss modulation beyond 3.8 μm . Moreover, this is the first fabricated electro-optic device on the lithium-niobate-on-sapphire platform, establishing a solid foundation for future developments.

Change in the manuscript:

- *“Alternative approaches such as external free-space Stark-effect modulators have achieved comparable bandwidths [20–22] and extinction ratios up to 3 dB [23]. These devices are particularly attractive because they rely on a refractive modulation mechanism rather than carrier absorption, enabling easier free-space coupling and potentially lower insertion losses. However, their scalability remains limited due to the high alignment sensitivity inherent to free-space configurations. In the integrated domain, III-V-based and group-IV carrier-injection or depletion modulators [24] can operate at longer wavelengths (up to 8 μm) but generally exhibit high optical losses. Despite their impressively low voltage-length product ($V\pi L$), these devices suffer from significant free-carrier absorption. Recent demonstrations include implementations on Ge-on-Si platforms [24] and silicon-on-insulator structures [25], which exhibit relatively high propagation losses due to lattice-mismatch defects between Ge and Si, and absorption in the silicon dioxide substrate, respectively. A newer technology relies on graded-index Ge-on-Si designs [26], which minimize defects and thus reduce losses, but still suffer from shallow modulation depths despite being very*

promising in terms of wavelength accessibility. Hybrid Si/LN transfer-printed modulators [27, 28] have extended lithium niobate operation into the mid-infrared, demonstrating good performance near 3.8 μm and showing promise for integrated spectroscopic sensing. However, fabrication complexity and limited electro-optic overlap still constrain their performance. Finally, lithium niobate on insulator (LNOI) modulators, despite their proven success in the near-infrared, are intrinsically limited to operation below approximately 3.8 μm [29] due to strong absorption in the buried SiO₂ layer [30].”

Below are detailed comments:

1. Title clarity:

Since a mid-IR modulator based on TFLN has already been demonstrated in the following paper, the title should be more specific, explicitly indicating that the work is based on the LNOS platform:

Hyeon Hwang, Kiyong Ko, et al., “A wide-spectrum mid-infrared electro-optic intensity modulator employing a two-point coupled lithium niobate racetrack resonator,” APL Photonics 10, 016116 (2025).

Response: We thank the reviewer for this valuable recommendation and apologize for having initially omitted this important reference. **The work by Hwang et al. (APL Photonics 10, 016116 (2025)) has now been cited in the revised manuscript. The title has been changed accordingly.**

Change in the manuscript:

- “Thin film lithium niobate on sapphire for integrated mid-infrared modulator”
Part Title

- “Finally, lithium niobate on insulator (LNOI) modulators, despite their proven success in the near-infrared, are intrinsically limited to operation below approximately 3.8 μm [29] due to strong absorption in the buried SiO₂ layer [30].” Part Main

2. Overstated claim:

The sentence on page 4 —

“To our knowledge, this represents the first photonic integrated MZM demonstrated in the MIR, surpassing previous demonstrated modulators in terms of speed and extinction ratio by nearly two orders of magnitude.”

— should be removed, as it is not accurate in light of prior demonstrations.

Response: Thank you for the observation, that is correct. The sentence has been revised for accuracy. As previously mentioned, earlier demonstrations relied on LNOI, carrier-depletion mechanisms or Si/LN hybrid platforms, which, while functional,

exhibit significant drawbacks such as high optical losses. **The text has been modified accordingly to clarify this distinction.**

Change in the manuscript:

- “Furthermore, we demonstrate full π -phase modulation in the MIR, uniquely combining integration, low propagation loss, extinction ratio and high-speed operation, representing a key milestone for MIR photonics. These results establish a pathway toward energy-efficient, high-performance mid-infrared photonic systems for applications in telecommunications, sensing, and quantum technologies.” Part Abstract

- Deleted the overstated claim on page 4

3. Insufficient references for MIR relevance:

The following statement is correct but lacks supporting citations:

“In this context, phase and amplitude modulation are essential techniques for MIR spectroscopy – improving sensitivity, resolution, and signal-to-noise ratio.”

Please include appropriate references that emphasize the importance of MIR modulation and its applications.

Response: We have now included several references that highlight the critical role of amplitude and phase modulation in enhancing the performance of mid-infrared spectroscopic systems.

Change in the manuscript:

- “In this context, efficient electro-optic modulators play a crucial role in both domains. External modulators, in particular, enable high modulation depths and can handle significantly higher optical power than directly modulated sources, making them highly advantageous for achieving strong, high-quality modulated signals in the mid-infrared.[8] The MIR spectral range also offers several benefits for spectroscopy, whereby precise control over the phase and amplitude of mid-infrared light is required. Amplitude modulation enables lock-in detection to improve sensitivity and suppress background signals [9, 10], and it can also be used to probe very fast phenomenon [11] and delicate samples [12]. For example, Tamamitsu et al. use a 0.1 ns (10 GHz) MIR pulse to confine heat diffusion and enhance photothermal efficiency, although it is currently generated by a bulky OPO system. Phase modulation, on the other hand, enables wavelength modulation spectroscopy [12] and the generation of frequency-comb light sources[13], providing powerful tools for high-resolution and broadband spectroscopy” – Part Main

4. Spectral limitation and platform comparison:

The experiments were conducted only between 3.9–4.1 μm , covering a narrow range with limited molecular fingerprint information. To better demonstrate the potential broadband applicability of the platform, I suggest adding theoretical loss calculations for both LNOI and LNOS platforms across the 3–5 μm range (e.g., in Fig. 1). Additionally, including Q-factor measurements from ring resonators would help substantiate the claim of achieving a low-loss MIR platform.

Response: Our experiments were conducted in the 3.9–4.5 μm range, which corresponds to the spectral region accessible with our available lasers. This range already extends beyond the limit of LNOI platforms (typically < 3.8 μm) with very important molecules fingerprint (such as CO₂, CO and N₂O). Particularly, the strong absorption band of CO₂ around 4.26 μm and Co at ~4.5 μm as well as the strong absorption of N₂O around 4.5 μm is very important. (<https://hitran.org/>)

[editorial note: third party material redacted]

Vainio, Markku, and Lauri Halonen. *Physical Chemistry Chemical Physics* 18.6 (2016): 4266-4294.

We chose to focus on this wavelength because it lies between two major absorption bands while remaining within the atmospheric transparency window, making it a good demonstration for applications in both free-space communication and spectroscopy.

Regarding the difference in loss between lithium niobate on sapphire and on silicon, we have added a reference in our response (see Fig. 1 in this document), which is also cited in the main text. As already mentioned, using thin-film lithium niobate on sapphire can reduce the loss at 4 μm by at least one order of magnitude. These results were obtained in this paper (Mishra, Jatadhari, et al. *Optica* 8.6 (2021): 921-924.) by evaluating the absorption loss for silica and sapphire bottom claddings with identical thin-film geometries, based on bulk absorption data and mode overlap calculations with the cladding.

We fully agree that Q-factor measurements would offer valuable insight into the intrinsic propagation losses. However, ring resonator characterization in the mid-infrared remains technically challenging due to the limited wavelength tunability of

available external cavity lasers, whose step size is too large to finely resolve narrow resonances. **Instead, we have included new cutback measurements, employing a grating coupler, which confirm low-loss waveguide performance of approximately 1.3 dB/cm. A plot showing the waveguide loss as a function of wavelength has also been added (Fig. 3d and Fig. C3c), together with additional details on the mid-infrared grating coupler.** Overall, these additional experimental and simulated results further support the broadband, low-loss potential of the LNOS platform for high-speed MIR electro-optic applications.

Change in the manuscript: Fig 3.d (Top left), Fig C3c (Top right) and Fig C3b (Bottom left)

5. Inconsistency in extinction ratio data:

In Fig. 3b, the extinction ratio (ER) is labeled as 34.1 dB, but the data suggest a minimum transmission of ~ 0.1 (corresponding to ~ 10 dB ER). The same issue appears in Fig. E5. Please verify and clarify this discrepancy. It may also be more appropriate to plot the modulation data on a logarithmic (dB) scale to better visualize the claimed high ER performance.

Response: The apparent discrepancy arises because the extinction ratio in Fig. 3b (and Fig. E5) was evaluated relative to the noise floor (shown as a dashed line in the plots), rather than solely from the visible signal minimum. The minimum transmission observed in linear scale (~ 0.1) does not reflect the true extinction-limited value, as the correct evaluation must be made with respect to the detector's noise floor (noise from

the detector without laser illumination). **We have now modified the figure accordingly in the manuscript to plot it in dB and better represent the value of the extinction ratio. (Fig. 3b)**

Change in the manuscript: Fig 3b

b

6. Missing references for wavelength-dependent modulation:

The following statement on page 9 should be supported by experimental references:

“This behaviour arises from a fundamental limitation — the Pockels effect induces a change in refractive index Δn , but the resulting phase shift decreases at longer wavelengths, as described by:”

I recommend citing the following works:

o Hixin Xue et al., “xue,” *Optica* 10, 125–126 (2023).

o Hyeon Hwang, Kiyong Ko, et al., “A wide-spectrum mid-infrared electro-optic intensity modulator employing a two-point coupled lithium niobate racetrack resonator,” *APL Photonics* 10, 016116 (2025).

Response: We respectfully disagree that additional experimental citations are required for this statement. The sentence describes a fundamental property of the Pockels effect, namely that the induced phase shift scales inversely with wavelength due to the dependence of the optical phase. This relationship arises directly from the standard electro-optic phase modulation expression and does not depend on specific experimental demonstrations. The provided formula already conveys this well-established physical dependence in classical electro-optic theory. **However, in the revised manuscript, we have clarified the purpose of the equation and more clearly**

emphasized the aspect we intended to illustrate, while also including the first reference you cited to show the increasing trend.

Change in the manuscript:

- *“Consequently, increasing the wavelength by a factor of approximately 2.5 results in a 3–4× increase in the intrinsic $V\pi L$ value.[46]”*

7. Discussion of the main limitation (high $V\pi$):

As discussed in the manuscript, the major limitation of the current platform is its high half-wave voltage ($V\pi$). It would be helpful to include additional information in the Supplementary Material regarding possible improvement limits through electrode design optimization. From a practical standpoint, it appears that the achievable enhancement may already be constrained by the inherent characteristics of MIR wavelengths.

Response: We agree that the relatively high half-wave voltage currently represents the main limitation of our platform.

As correctly noted, the achievable enhancement is indeed constrained by the intrinsic characteristics of mid-infrared operation. The longer wavelengths require wider optical modes and larger electrode spacing to minimize absorption losses, which intrinsically increases $V\pi$. Nevertheless, several trade-offs can be considered. For example, reducing the electrode spacing would lower $V\pi$ but at the cost of higher optical loss, while operating at shorter wavelengths would allow tighter electrode confinement and improved modulation efficiency. **To complement this analysis, we have performed additional experimental evaluations of the losses as a function of the spacing between the waveguide and the electro-optic electrodes. (Fig 3c & Fig. C3.d)**

Furthermore, alternative device architectures are mentioned in the revised manuscript and could be employed to reduce the effective driving voltage. In particular, resonant configurations such as ring-assisted Mach–Zehnder modulators (e.g., Xue, Yu, et al., “Breaking the bandwidth limit of a high-quality-factor ring modulator based on thin-film lithium niobate,” *Optica* 9, 1131–1137 (2022)) can significantly lower $V\pi$ while maintaining compact device footprints. This approach will be the focus of future developments on our platform.

Change in the manuscript:

Overall evaluation:

This is a well-executed and potentially valuable study demonstrating MIR modulation on an LNOS platform. However, in its current form, the manuscript lacks sufficient distinction from prior work and requires additional data, clarifications, and references to convincingly justify publication in Nature Communications.

We thank the reviewer for their valuable input and hope that, with the added material and revisions, we have addressed the concerns and strengthened the manuscript for publication.

Reviewer #3 (Remarks to the Author):

In the manuscript "Integrated thin film lithium niobate mid-infrared modulator", the authors report a Mach-Zehnder electro-optic modulator (MZM) on lithium niobate on sapphire (LNOS) nanophotonic platform, using which they demonstrate phase and amplitude modulation in the mid-infrared (MIR), around 4-microns wavelength. They further demonstrate data-transmission and generation of an electro-optic (EO) comb in this spectral region.

Upon reading this manuscript, I have the following questions and feedback for the authors:

- Would it be possible for the authors to discuss in more detail the design considerations of their device, perhaps in section 2.1, or in a separate section in the appendix? I believe it would be helpful for the reader to understand how key figures of merit, such as the $V_{\pi}L$ and EO bandwidth, are dependent on the various design parameters, why particular waveguide dimensions and electrode dimensions were chosen, and if special attention was paid to group velocity matching and impedance matching. While some of these design considerations and simulations are interspersed with results, I think it'd be beneficial to have most of them collected in one place to

provide a full picture of designing an MZM in the MIR, and to understand if any design optima may differ from those that are well-established in the near infrared.

Response: While some design considerations were already included in the original manuscript, we fully agree that a more comprehensive overview would benefit the reader. **We have therefore expanded the description and added complementary details in the Appendix C and E. We provided an analysis of the losses as a function of the spacing between the electro-optic electrode (Fig.3c top Panel & Fig. C3d). We have also included the simulations performed and discussions on the Appendix E on RF design of the electrode. It shows how we designed and verify that the electro-optic electrodes were impedance-matched to 50Ω (Fig. E5a) and that the electrical S_{11} measurements exhibited minimal reflections up to 20 GHz (Fig. E5b).**

Change in the manuscript: Captions and discussions linked to those figures:

- The authors use a MIR-compatible multimode lensed fiber to couple light into the input waveguide. How do they ensure that they only have fundamental mode excitation in the waveguide?

Response: The waveguide used in our experiments is designed to support only the fundamental TE/TM mode at the operating wavelength. Therefore, even though a multimode lensed fiber is used for coupling, only the fundamental mode of the waveguide can propagate, and higher-order modes. TE mode can then be differentiated

from the TM mode based on the modulation strength of the modulator, the different polarizations use different elements of the LN Pockels tensor, with the TE mode being modulated roughly three times more strongly than the TM mode.

- Fig 3d has only four wavelengths characterized between 3.95 and 4.3 microns. Based on the QCL spectrum in appendix B, would it be possible to have measurements at more wavelengths in this range? It would be good to understand the apparent dispersive trends and ascertain if they arise from experimental noise or if they correspond to oscillations in the device efficiency.

Response: The observed increase of V_π with wavelength primarily arises from the induced phase shift decreases with increasing wavelength, consistent with the expected wavelength dependence of the electro-optic effect. In addition, it also comes from the waveguide design, which is optimized for operation around 4.0 μm . As the wavelength increases beyond this point, the optical mode becomes less confined in the lithium niobate layer and more in the sapphire substrate, leading to reduced modulation efficiency and a corresponding increase in V_π . **We have added a simulation in Fig. 3d (top figure) that evaluates the variation of $V_\pi \cdot L$ as a function of wavelength. The simulated results show good agreement with the experimental data.**

In the revised manuscript, we have also extended the characterization up to 4.5 μm . The measurement over the 3.9–4.5 μm wavelength range remains constrained by the discrete emission modes of the available DFB QCLs (4.3 μm and 4.5 μm) and by the limited tuning bandwidth of our external-cavity QCL sources around 4 μm , which do not provide single-mode operation between 4.1 μm and 4.3 μm .

Change in the manuscript:

- The generation of MIR EO comb through the MZM is certainly of great interest due to its potential applications, however this portion of the manuscript appears to be lacking in some details, such as the driving voltage and observed bandwidth, as well as in

discussions about ways to improve the bandwidth of the comb. Also, can multiple devices (varying waveguide thickness, electrode length, etc.) be tested for EO comb generation and the results compared?

Response: In the revised manuscript, we have included a more complete description of the measurement conditions and the MIR electro-optic comb generation at different driving frequencies. The device used for comb generation is the one exhibiting the lowest V_{π} , which is the most relevant parameter for efficient comb formation. The waveguide thickness has already been optimized to achieve low propagation loss, strong optical confinement, and single-mode operation for the TE polarization. Nevertheless, in future developments, the device length could be slightly increased (relatively to the size of the chip) to further enhance the modulation efficiency and broaden the comb bandwidth.

Change in manuscript:

d

- “Using our modulator, we successfully generated an optical frequency comb centered around the carrier frequency for three different modulation frequencies, as measured by the OSA with a resolution of 1.9 GHz. As shown in Fig. 4d, the comb is centered at a wavelength of approximately $4 \mu\text{m}$, with symmetric sidebands clearly resolved, confirming the expected modulation behavior. At 7 GHz, based on the relative powers of the carrier and higher-order sidebands, we observed a maximum attenuation of 6.4 dB per sideband peak, leading to a spectral bandwidth of around 70 GHz, which reflects efficient energy transfer

from the central mode into the sidebands. The energy transfer, highlighted by the slope of the comb, decreases with increasing frequency at 10 GHz and 12 GHz, which we attribute to the combined effects of a higher effective $V\pi L$ at elevated frequencies and reduced electrical power delivery to the electrodes due to impedance mismatch. The measured spectrum is not limited by the OSA noise floor, which is around -70 dBm. In fact, a broader comb could be obtained using a laser source with lower intrinsic amplitude noise. The relatively high noise level, around -50 dBm in our measurements, arises from the use of the EC-QCL, which typically exhibits higher amplitude and frequency noise due to mechanical tuning elements and longer cavity lengths. For practical spectroscopic applications, a low-noise source such as a $4\ \mu\text{m}$ DFB QCL or an interband cascade laser would significantly enhance the signal-to-noise ratio and allow observation of broader comb spectra, with noise down to -60 dBm and bandwidths exceeding 100 GHz. The ability to tune the spacing between comb lines is a powerful tool for scanning absorption bands. These results demonstrate the modulator's capability to generate rich frequency content through deep amplitude modulation, confirming its potential as a building block for mid-infrared frequency-comb generation and broadband spectroscopic systems."

- From the transcript: "While the simulation shows good agreement with the experimental results for the $1.5\ \mu\text{m}$ devices, a slight discrepancy is observed for the $0.9\ \mu\text{m}$ case, likely due to a non-uniform gap length along the electrode."; why would the gap between lithographically defined electrodes be nonuniform? Could it be due to improper metal liftoff? An optical microscope observation might provide some clarification.

Response: We re-examined the electrode spacing with an SEM between the waveguide and the metal electrode, and the measurements confirmed that the spacing is uniform and consistent with the design within ± 100 nm. Therefore, the previously stated hypothesis regarding a non-uniform gap was incorrect. **The slight discrepancy between the simulation and the experimental results remains unexplained at this stage, and the corresponding sentence has been removed from the revised manuscript.**

- Based on the propagation loss estimates from the cutback measurements, can the authors ascertain the excess loss through the MZM and if / how it depends on the waveguide and electrode designs?

Response: In the revised manuscript, we have included new data obtained from cutback measurements using grating coupler structures, which allow a more precise evaluation of the waveguide propagation loss independent of coupling variations. (Fig 3d, FigC3b, FigC3c). These additional results confirm the low-loss performance of the passive waveguides on the LNOS platform.

As already mentioned, we have added a new figure analyzing the additional loss induced by the electrodes as a function of electrode spacing and length. (Fig3c, Fig C3d). This provides a direct estimate of the excess loss in the MZM and illustrates the trade-off between modulation efficiency and optical absorption from the metal electrodes.

Changes in the manuscript:

Some minor comments:

- It appears from Fig. 1b that the waveguide width is 4 microns, while I believe the intention there is to show the wavelength?

Response: The value indicated in Fig. 1b (4 μm) indeed refers to the waveguide width, not the wavelength. **We have clarified this point by changing the scale with a length of 2 μm to avoid confusion with the wavelength.**

Changes in the manuscript:

- The layout of the electro-optic electrode is not clear to me from figure 1d; perhaps the authors could change the color / contrast of the image to make the photonics components more visible in this image?

Response: The contrast and color scheme of Fig. 1d have been improved in the revised manuscript to make the electrode layout and photonic components more clearly visible.

Change in the manuscript:

- Typo in figure 3b caption, V_π of 28 V (not Vcm)?

Response: The change has been implemented in the revised manuscript.

- Figure 3c caption needs some clarification. I would also recommend that the authors plot the simulations as solid / dashed lines and reserve cross / dot for experimental data points.

Response: The change has been implemented in the revised manuscript.

Change in the manuscript:

- Captions for Fig.4c&d appear to be interchanged.

Response: The captions for Figs. 4c and 4d were indeed interchanged in the original submission. This has been corrected in the revised manuscript.

- Fig. B2 caption incorrect.

Response: The caption for Fig. B2 has been corrected in the revised manuscript.

- Please clarify the TOPS set point for the EO measurements.

Response: The set point corresponds to the condition where the optical output power of the interferometer is half of its maximum value, ensuring operation at the quadrature point for optimal electro-optic modulation efficiency. This condition was verified either by monitoring the static transmission curve or by observing the modulator output on an oscilloscope and confirming symmetric modulation behavior with respect to the applied voltage. **We have clarified the method used to set the device at the quadrature point.**

Change in the manuscript: “The TOPS voltage set point corresponds to the condition whereby the optical output power of the interferometer is half of its maximum value, ensuring operation at the quadrature point for optimal electro-optic modulation efficiency. This condition was also verified by observing the modulator output on an oscilloscope and confirming symmetric modulation behavior with respect to the applied voltage.”

Overall, my impression is that while the authors have adequately demonstrated MZM in the LNOS platform in the MIR, if they could expand upon their demonstration and / or analysis of MIR EO-comb generation in this platform, it would substantially increase the contribution of their work and make it more suitable for publication in Nature Communications.

Response: We thank the reviewer for this thoughtful comment and for recognizing the significance of our demonstration. We fully agree that extending this work toward optimized mid-infrared electro-optic comb generation on the LNOS platform would represent an exciting and impactful next step.

However, the main objective of the present work is to establish and experimentally validate the fundamental operating principle of the LNOS platform by demonstrating a high-efficiency Mach–Zehnder modulator operating in the mid-infrared with a low-loss platform. In this paper, we focus on showcasing the high-performance fabrication process, including the realization of deep etching ($\sim 1 \mu\text{m}$) on a transparent sapphire substrate and the successful integration of electro-optic electrodes and thermal phase shifters on a state-of-the-art low-loss platform with propagation losses around 1.3 dB/cm @ 4 μm . These results represent an important milestone, confirming the feasibility of implementing complex active photonic components on this emerging platform, while also enabling nonlinear processes such as spontaneous parametric down-conversion and difference-frequency generation in the mid-infrared.

We fully acknowledge the reviewer’s suggestion regarding EO-comb generation. Indeed, resonant and non-resonant comb architectures are currently under active investigation in our group, and preliminary results have already been obtained. Nevertheless, demonstrating and analyzing optimized EO-comb generation would require a substantially different device design and extensive discussion of cavity dynamics and

dispersion engineering, which would go beyond the scope of this paper. We have nevertheless included the best possible comb that can be generated with this device architecture to demonstrate the potential of this platform in this regard.

We therefore believe that the current manuscript appropriately focuses on demonstrating the core capability and fabrication maturity of the LNOS platform, which lays the essential groundwork for future demonstrations of EO-comb generation and other advanced functionalities in the mid-infrared.

Response to the Reviewers' comments on the manuscript titled "Thin film lithium niobate on sapphire for integrated mid-infrared modulator" submitted to Nature Communications

Thank you very much for the opportunity to revise our manuscript. We would like to express our sincere gratitude to the Editor and Reviewers for their evaluation of our work, which has greatly improved its quality.

Below, we provide a detailed, point-by-point response to the suggestions and questions raised by the Reviewers. Along with this resubmission, we have included a revised version of the manuscript that incorporates these constructive comments highlighted in green.

Reviewer #1 (Remarks to the Author):

The authors have adequately addressed my previous comments. Regarding the revised manuscript, I would like to suggest one minor improvement. To better illustrate the experimental configuration for the free-space transmission at 10 Gbit/s over a 0.5 m distance, the addition of a figure showing a photograph or a link diagram of the actual setup would be valuable. Such a depiction would greatly aid in the interpretation and visualization of this specific experimental scenario.

Response: Thank you for the positive assessment. Based on your suggestion, we have now implemented a schematic of the actual setup and included it in the revised manuscript in **Fig 4.c** where we added a schematic representation of the free space transmission setup on the top panel. This depiction should make the experimental

arrangement clearer and help readers visualize the specific free-space transmission scenario.

Reviewer #2 (Remarks to the Author):

I appreciate the authors' detailed response and the revisions made to the manuscript. I understand the authors' position that a full demonstration of spectroscopy applications is beyond the scope of this work and that the manuscript aims to serve as a "foundational demonstration" of the LNOS platform in the mid-infrared.

However, if the focus is shifted towards the fundamental aspects of the platform rather than practical applications, the analysis of the optical loss mechanisms must be significantly more rigorous.

Specifically, I strongly suggest adding a fundamental analysis comparing the experimental losses with the theoretical limits. For instance, based on Mishra, Jatadhari, et al., *Optica* 8.6 (2021): 921-924, the calculated absorption loss of the LNOS platform is approximately ~ 0.01 dB/cm at $3.5 \mu\text{m}$ and ~ 0.1 dB/cm at $4 \mu\text{m}$.

In contrast, the propagation losses reported in this manuscript (e.g., ~ 1.3 dB/cm at $4 \mu\text{m}$) are significantly higher than these theoretical limits. This discrepancy suggests that the current device does not fully exploit the intrinsic low-loss advantage of the LNOS platform. In fact, such loss values might be comparable to what could be achieved on LNOI platforms in certain conditions, thereby weakening the argument for LNOS.

Therefore, to justify the significance of this work as a "foundational demonstration," the authors must:

1. Explicitly compare their measured losses with the theoretical absorption limits calculated in literature (e.g., Mishra et al.).
2. Provide a detailed analysis of the sources of the excess loss (e.g., scattering due to sidewall roughness vs. material absorption).
3. Discuss why the current experimental values deviate from the theoretical potential and how this gap can be bridged in future work.

Response: We thank the reviewer for the thoughtful evaluation and for clearly understanding the purpose of the article. We agree that a detailed loss study is indeed very valuable, especially for a foundational demonstration of the LNOS platform.

As the reviewer notes, theoretical absorption limits represent the ultimate bound on material performance. However, in practical integrated photonic systems, losses are almost always dominated by fabrication-related imperfections rather than intrinsic material absorption. Even on the well-established LNOI platform, the theoretical loss limits have not yet been experimentally achieved. As shown in the referenced work, the lowest loss demonstrated is around 1.3 dB/m at 1550 nm, whereas the theoretical limit

from the material is evaluated to be 0.1 dB/m, an order of magnitude lower (Zhu X et al., “Twenty-nine million intrinsic Q-factor monolithic microresonators on thin-film lithium niobate”, *Photonics Research*, 2024).

In our case, two primary mechanisms contribute to the excess loss:

- Imperfections at the bonded interface between sapphire and lithium niobate. Residual stress or interfacial defects can introduce additional scattering or localized absorption that increases propagation loss. Because the thin-film lithium niobate we use is commercially supplied, we have no control or knowledge over its bonding process, making it difficult to directly verify the interface quality.
- Sidewall roughness arising from the lithography process, which is transferred during the dry-etching step, also contributes to propagation loss. A representative SEM image of our waveguide is included just below, showing the sidewall roughness, although additional process refinement is expected to reduce it even further.

We also appreciate the reviewer’s comment suggesting that LNOI might achieve losses comparable to LNOS at 4 µm. To our knowledge, published LNOI waveguides operating beyond 3.5 µm generally report losses significantly above 1 dB/cm, primarily due to absorption in the SiO₂ BOX layer. To the best of our knowledge there is no demonstration showing lower loss at 4 µm for LNOI.

For completeness, simulated material absorption in lithium niobate at 4 µm is below 0.1 dB/cm, whereas absorption in silica for a thin film of 1.5 µm thickness is near 1

dB/cm. However, as already mentioned, experimental losses in thin-film lithium niobate often differ significantly from theoretical predictions.

A relevant reference for mid-infrared LNOI loss is *Hwang H et al.*, “A wide spectrum mid-infrared electro-optic intensity modulator employing a two-point coupled lithium niobate racetrack resonator”, *APL Photonics*, 2025. This paper reports simulated LNOI losses near 1 dB/cm for single-mode operation, whereas the experimental extraction based on the intrinsic quality factor yields approximately 4 dB/cm at 3.8 μm , despite the more mature LNOI fabrication. The authors also note in this paper, consistent with our observation, that “the optical losses from the bottom SiO_2 layer and the electrode increase with wavelength”. Consequently, one expects the loss to continue increasing beyond 3.8 μm . To conclude, although LNOS fabrication is far less mature, the loss on our platform is likely to be reduced by roughly 3 dB/cm compared to LNOI.

Finally, increasing the waveguide width would reduce the propagation loss although this comes at the cost of entering a multimode regime. Indeed, as shown in *Zhang M et al.*, “Monolithic ultra-high-Q lithium niobate microring resonator”, *Optica* 2017, the loss decreases by roughly one order of magnitude when the structure transitions from a single mode waveguide to a multimode one as it weakens the interaction between the optical mode and the lossy boundaries of the waveguide.

We thank the reviewer again for raising these important points. We have added discussions and clarified the distinction between intrinsic material limits and fabrication-related constraints in this inaugural demonstration of the LNOS platform.

Changes in the manuscript: “The losses here are dominated by sidewall roughness and could be further reduced by improved fabrication techniques, as theoretical studies show that losses could be reduced to as low as 0.1 dB/cm [40]. Moreover, moving to a multimode waveguide geometry would allow these losses to be reduced even further because the optical mode becomes less sensitive to scattering at the waveguide boundaries [44]”

Reviewer #3 (Remarks to the Author):

The authors have adequately addressed my queries and improved the manuscript.

Response: We thank the reviewer for the positive assessment. We appreciate the helpful feedback which has significantly improved the manuscript. Thank you again for the thorough and constructive review.

Response to the Reviewers' comments on the manuscript titled "Thin film lithium niobate on sapphire for integrated mid-infrared modulator" submitted to Nature Communications

Thank you for the opportunity to revise our manuscript. Below, we provide a detailed response to the suggestions and questions raised by the Reviewer, clarifying the feasibility of the requested analysis. Along with this resubmission, we have included a revised version of the manuscript with changes highlighted in purple.

We would first like to emphasize that the demonstrated platform already exhibits low propagation loss for state-of-the-art single mode mid-infrared waveguides, particularly considering the long operating wavelength and the strong mode confinement required for single mode operation. As discussed in the manuscript, propagation losses could be further reduced by moving to a multimode waveguide configuration. However, such an approach would not be relevant for the device demonstrated here, where single mode operation is essential for efficient modulation.

Regarding the question of accurately calculating or isolating propagation loss mechanisms, we believe it is important to distinguish between the physical origin of the losses and the practical limitations of the available measurements.

First, any loss originating from the lithium niobate and sapphire interface introduced during the wafer level bonding process is not under our direct control, as the detailed process flow and the associated material and interface imperfections are not fully accessible to us. Since we rely on commercial wafer manufacturers, crystallographic defects or interfacial layers must be accepted as intrinsic loss factors contributing to degraded performance. These contributions are very difficult to isolate experimentally or to model quantitatively in a predictive manner.

Second, from a modeling perspective, losses associated with waveguide sidewall roughness are well understood to arise from Rayleigh scattering, with a characteristic wavelength dependence scaling as λ^{-4} . This wavelength scaling is well established in the literature, including the references cited by the reviewer, and provides a sound qualitative physical framework for interpreting the observed trends. However, converting this scaling law into an accurate prediction of the absolute loss magnitude, that could be included in the paper, would require detailed knowledge of sidewall roughness statistics, correlation lengths, thin film thickness variation, and fabrication specific parameters. While we could provide an estimate of the scattering loss and include it in the manuscript, we are concerned that such a result would not be truly representative of the actual losses.

In thin film lithium niobate on silicon photonics, remarkably low propagation losses have been demonstrated due to precise control of lithography and etching processes,

particularly through argon ion milling. This technique has a smoothing character that enables sub nanometer sidewall roughness. In contrast, our longer operating wavelength requires significantly deeper etching into the film. This requirement, combined with the transparent and insulating nature of our wafer stack, necessitates a different and more complex lithography approach, which in turn leads to increased waveguide sidewall roughness. For these reasons, we attribute the dominant loss mechanism in our devices to scattering, with an additional contribution arising from interfacial defects at the lithium niobate sapphire boundary.

From an experimental standpoint, the type of conclusive evidence requested by the reviewer would require extremely high precision propagation loss measurements together with very high fabrication reproducibility. In practice, the most accurate loss characterization is achieved using ring resonators and quality factor analysis. Such measurements are not currently accessible due to the lack of a suitable finely tunable narrow linewidth source. Moreover, a simple wavelength sweep would not yield conclusive trends for this type of analysis. As the wavelength decreases, the effective mode index increases and the effective mode area shrinks, resulting in reduced modal overlap with the waveguide sidewalls and therefore a decrease in the wavelength dependence of the scattering cross section. A rigorous investigation would thus require simultaneous variation of both wavelength and waveguide cross section, an effort that we believe lies beyond the scope of this work. Another possible method would be to tune the resonance using a thermal phase shifter placed close to the ring. This approach was attempted but yielded unsuccessful results due to crosstalk from the thermal phase shifter, leading to nearly identical extracted values that were not conclusive enough to be presented in a publication.

Finally, due to the reasons explained above, propagation losses were extracted using a cutback method. While this provides a robust estimate of the average loss, it lacks the precision required for a detailed decomposition of individual loss mechanisms. Unavoidable variations in coupling efficiency arising from differences in alignment quality, facet quality, or grating couplers between waveguides introduce uncertainties that limit the sensitivity of this method for the refined loss analysis being requested.

The only additional experimental approach that would be realistically accessible is a systematic study of propagation loss as a function of waveguide width, to evaluate the overall lowest achievable loss. While this could provide indirect insight into scattering related losses, it would require dedicated design, fabrication, and characterization of a large set of structures, and would constitute a substantial new section focused specifically on waveguide loss engineering. Such a study would also be only weakly related to the present manuscript, as the waveguide presented here is strictly single mode. We believe such a study would go beyond the scope of a standard revision and

would more naturally form the basis of a separate, focused publication on propagation losses in this platform.

Refining the fabrication process is currently the focus of our efforts, with the goal of achieving waveguide quality comparable to that routinely demonstrated in lithium niobate photonics on silicon at visible and near infrared wavelengths. Moreover, we believe that replotting the simulated material loss of lithium niobate on sapphire compared to lithium niobate on silicon, as already presented in the referenced work by Mishra et al. (Optica 2021), would not constitute a scientific novelty.

As discussed by the reviewer, we agree that attributing the losses to a specific origin without ambiguity requires additional investigation. For this reason, we have rephrased the sentence regarding the origin of the losses in the manuscript, highlighted in purple. In addition, we have cited the papers mentioned by the reviewer. The revised sentence now reads:

“The losses are hypothesized to be dominated by sidewall roughness [44, 45] and could be further reduced through improved fabrication techniques, as theoretical studies suggest that losses approaching 0.1 dB/cm may be achievable [40]. Moreover, adopting a multimode waveguide geometry could further reduce these losses, since the optical mode becomes less sensitive to scattering at the waveguide boundaries, although this would reduce the efficiency of the presented modulator [46]”

We hope this clarifies why a fully predictive loss calculation or a conclusive experimental isolation of scattering loss would represent a significant undertaking. While such an investigation is interesting and relevant on its own, we believe that it goes beyond the scope of demonstrating the capabilities of our modulator on this platform. We thank the editor and the reviewers again for all their comments and for helping to improve the quality of the manuscript.